# AQuA: Toward Strategic Response Generation for Ambiguous Visual Questions

**Jihyoung Jang**[1]    **Hyounghun Kim**[1,2]
[1]Graduate School of Artificial Intelligence, POSTECH
[2]Department of Computer Science and Engineering, POSTECH
{jihyoung, h.kim}@postech.ac.kr

## Abstract

Visual Question Answering (VQA) is a core task for evaluating the capabilities of Vision–Language Models (VLMs). Existing VQA benchmarks primarily feature clear and unambiguous image–question pairs, whereas real-world scenarios often involve varying degrees of ambiguity that require nuanced reasoning and context-appropriate response strategies. Although recent studies have begun to address ambiguity in VQA, they lack (1) a systematic categorization of ambiguity levels and (2) datasets and models that support strategy-aware responses. In this paper, we introduce **A**mbiguous Visual **Qu**estion **A**nswering (AQuA), a fine-grained dataset that classifies ambiguous VQA instances into four levels according to the nature and degree of ambiguity, along with the optimal response strategy for each case. Our evaluation of diverse open-source and proprietary VLMs shows that most models fail to adapt their strategy to the ambiguity type, frequently producing overconfident answers rather than seeking clarification or acknowledging uncertainty. To address this challenge, we fine-tune VLMs on AQuA, enabling them to adaptively choose among multiple response strategies, such as directly answering, inferring intent from contextual cues, listing plausible alternatives, or requesting clarification. VLMs trained on AQuA achieve strategic response generation for ambiguous VQA, demonstrating the ability to recognize ambiguity, manage uncertainty, and respond with context-appropriate strategies, while outperforming both open-source and closed-source baselines.[1]

## 1 Introduction

Recent advances in Vision–Language Models (VLMs) (Dai et al., 2023; Liu et al., 2023; Chen et al., 2024; Bai et al., 2025) have significantly improved their performance across a broad range of Visual Question Answering (VQA) tasks (Antol et al., 2015; Goyal et al., 2017; Gurari et al., 2018; Singh et al., 2019; Mathew et al., 2021). Traditional VQA benchmarks primarily evaluate whether models can provide correct answers to clearly stated, unambiguous questions paired with well-defined images (Johnson et al., 2017; Hudson & Manning, 2019; Biten et al., 2019). While such benchmarks are valuable for assessing basic multimodal reasoning, they fail to capture a critical challenge in real-world use: the ability to handle ambiguous or unclear queries. This capability remains underexplored.

In human communication, ambiguity is typically resolved through contextual inference or follow-up questions. For example, when asked *"What brand is this vehicle?"* in an image with multiple cars, one may seek clarification or infer the intended car from context. Previous research in ambiguous VQA has mainly focused on making models always ask clarifying questions when uncertain (Jian et al., 2025). While this binary answer-or-ask strategy can be useful, it does not reflect real-world conversational dynamics, where clarification is not always the most efficient approach (Chen et al., 2025). Humans instead adapt their strategy to the situation—sometimes inferring intent from context, sometimes offering multiple plausible answers when they are few, and requesting clarification only when necessary.

---

[1]Our code, dataset, and model are publicly available at https://aqua-iclr2026.github.io/.

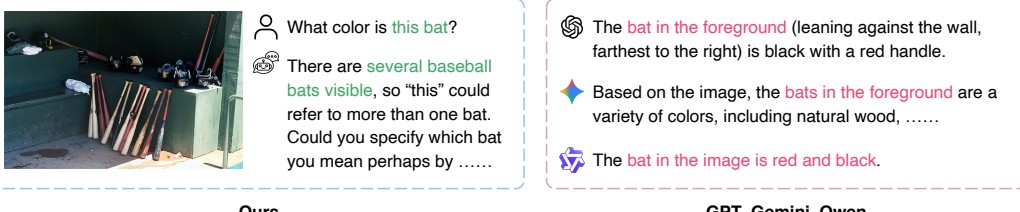

Figure 1: Examples of model responses to an ambiguous visual question. In this image, none of the bats is visually salient, making the visual context ambiguous. While GPT, Gemini, and Qwen provide answers by arbitrarily selecting (e.g., the bat in the foreground) despite the ambiguity, our model, which is trained to handle such cases strategically, requests clarification instead.

To bridge this gap, we propose **A**mbiguous Visual **Qu**estion **A**nswering (AQUA), a novel dataset designed to enable VLMs to choose contextually appropriate strategies for ambiguous VQA. Our dataset categorizes VQA instances into four fine-grained levels, based on both the nature and degree of ambiguity: (Level 0) unambiguous questions, (Level 1) questions whose intended referent can be inferred from context, (Level 2) questions with multiple plausible answers where listing options is preferable, and (Level 3) highly ambiguous questions requiring explicit clarification. To our best knowledge, AQUA is the first resource enabling systematic training and evaluation of strategy selection across these distinct ambiguity scenarios.

We empirically show that both open-source models (Bai et al., 2025; Chen et al., 2024) and high-performing closed-source models (GPT-5[2] and Gemini 2.5 Flash[3]) fail to properly handle ambiguous VQA, often responding overconfidently rather than adapting to the ambiguity (Figure 1). Building on these findings, we train models on AQUA to enable them to produce strategy-aware responses that adapt to the degree of ambiguity. Since generating strategy-adaptive responses is highly challenging for baseline models, we begin with supervised fine-tuning (SFT) to explicitly teach them the space of possible strategies. SFT provides a solid foundation for ambiguity-aware responses, but it does not directly optimize for strategic choice. To address this, we further apply Group Relative Policy Optimization (GRPO) (Shao et al., 2024), rewarding models when they produce strategy-aligned outputs and thereby improving their ability to adapt to varying degrees of ambiguity. VLMs fine-tuned on AQUA achieve substantially better performance by developing adaptive ambiguity-handling abilities. Our analysis demonstrates not only whether VLMs genuinely understand ambiguity and respond strategically, but also why such strategy-based responses are effective.

Our contributions in this paper are as follows:

1. We propose AQUA, a novel VQA dataset designed to train and evaluate how VLMs handle ambiguity. AQUA is organized into four fine-grained levels based on the degree and nature of ambiguity, enabling systematic analysis of response strategies across different ambiguous scenarios.
2. We fine-tune open-source models on AQUA, and they outperform larger open-source and high-performing closed-source models by autonomously selecting contextually appropriate response strategies.
3. Through extensive analysis, we verify why VLMs fail to generate strategic responses, analyze their error patterns, and confirm the effectiveness of strategic responses in handling ambiguity.

## 2 RELATED WORKS

**Ambiguity in Question Answering.** Traditional Question Answering (QA) benchmarks typically focus on unambiguous question–context pairs with clear answers, which effectively measure models' basic comprehension but fail to assess their ability to handle ambiguity (Rajpurkar et al., 2016; Joshi et al., 2017; Kwiatkowski et al., 2019). In text-based QA, ambiguity has been extensively studied (Min et al., 2020; Stelmakh et al., 2022; Kim et al., 2023; Lee et al., 2023; Li et al., 2025b), whereas visual QA research has only recently begun addressing this gap. For example, Focus Ambiguity (Chen et al., 2025) analyzes the responses of GPT-4o and InternVL2 to ambiguous questions, revealing that models often generate answers that appear plausible but lack semantic adequacy.

---

[2] https://openai.com/index/introducing-gpt-5/
[3] https://deepmind.google/models/gemini/flash/

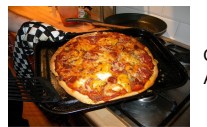

Q: What food is on the baking tray?
A: A pizza is on the baking tray.

**Level 0 (Unambiguous Questions)**

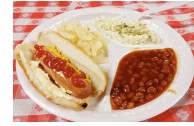

Q: What toppings are on this?
A: The term "this" points to the main item, the hot dog in the bun, and it's topped with mustard and ketchup.

**Level 1 (Low-Level Referential Ambiguity)**

Q: What is this player doing right now?
A: The reference might be to the Mariners fielder in white at first, ......, or to the runner in gray with a blue helmet and number 49, ......

**Level 2 (Multiple Valid Interpretations)**

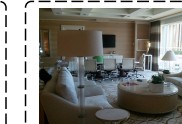

Q: What shape is this furniture?
A: Could you clarify which furniture you mean? There is a lot of furniture visible in the room visible in the room, so I can't tell which one's shape ......

**Level 3 (High-Level Ambiguity Requiring Clarification)**

Figure 2: Examples of the four ambiguity levels in AQUA.

ClearVQA (Jian et al., 2025) trains LLaVA to ask clarifying questions for ambiguous queries, but adopts a rigid binary strategy by always seeking clarification, without adapting to different types or degrees of ambiguity, which limits its practicality. In contrast, VAGUE (Nam et al., 2024) introduces a benchmark specifically designed to evaluate how visual contexts help resolve ambiguous linguistic expressions. To the best of our knowledge, AQUA is the first dataset to provide a fine-grained categorization of ambiguity in VQA, thus enabling systematic evaluation of diverse and context-appropriate response strategies.

**Uncertainty Handling Strategies.** While Large Language Models (LLMs) and VLMs can respond with "I don't know" in uncertain situations, they often show a tendency to answer even unanswerable questions (Guo et al., 2024; Li et al., 2025a). Previous research has primarily addressed this problem through binary approaches: training models to respond only when confident and to abstain when uncertain (Whitehead et al., 2022; Jian et al., 2025). These methods focus mainly on teaching models when to withhold responses. However, simply abstaining in uncertain situations does not always align with real-world human behavior (Liu et al., 2025). Depending on the degree of uncertainty, humans may leverage contextual clues to infer answers (Nam et al., 2024), provide all possible answers when there are only a few viable options, or ask follow-up questions to resolve ambiguity (Jian et al., 2025). We adopt this perspective in the context of ambiguous VQA, examining how VLMs should respond based on different types and degrees of ambiguity. To our knowledge, this is the first work that enables models to select among multiple response strategies based on specific ambiguous scenarios.

## 3 DATASET

In this work, we introduce **A**mbiguous Visual **Qu**estion **A**nswering (AQUA), a novel dataset that enables not only comprehensive evaluation but also effective training of VLMs on ambiguity in VQA. Unlike existing datasets that treat ambiguous queries in a uniform or binary fashion, our dataset systematically categorizes ambiguity into four distinct levels, enabling controlled and fine-grained training and evaluation.

### 3.1 LEVEL DEFINITIONS

In natural human communication, when confronted with ambiguous questions about visual information, people do not rely on a single strategy. Instead, they adapt their response according to the situation: asking for clarification when necessary, inferring answers from contextual cues when ambiguity is low, or enumerating all candidates when multiple plausible targets exist. For example, when looking at a crowded scene and asked, *"What is that person wearing?"*, a human might respond *"Which person?"* if there are several individuals, or directly answer if only one person is prominently visible.

Our goal in designing AQUA is not only to test whether VLMs can request clarification, but also to assess whether they can strategically respond using contextual reasoning when faced with ambiguity. To this end, we construct our VQA dataset with the following four levels (Figure 2):

- **Level 0. Unambiguous Questions**: Standard VQA cases with clear, unique answers, such as *"What food is on the baking tray?"* when there is only one tray with food. This category serves

as a control group to verify that models can still perform well on conventional VQA without over-applying ambiguity-handling strategies.

- **Level 1. Low-Level Referential Ambiguity**: Questions often involve pronouns like *"it"*, *"this"*, *"that"*, or *"these"* where context makes the referent obvious. For instance, in the example *"What toppings are on this?"*, the term *"this"* can be resolved from context because the hot dog is the only plausible referent for a topping-related question. Thus, the model is expected to infer that *"this"* refers to the hot dog and directly provide the corresponding answer, rather than treating the question as ambiguous.

- **Level 2. Multiple Valid Interpretations**: In these cases, offering all reasonable interpretations is preferable while asking for clarification may be unnecessary or inefficient. For example, consider the question *"What is this player doing right now?"* in an image where two baseball players are engaged in clearly distinct activities, with one running and the other fielding. At this level, there are only two or three plausible interpretations, and mentioning all of them is more efficient than asking for clarification.

- **Level 3. High-Level Ambiguity Requiring Clarification**: Questions that genuinely require clarification due to a high level of ambiguity. For example, in the question *"What shape is this furniture?"*, the image contains many visually similar objects, including multiple sofas, tables, desks, and lighting fixtures, making it unclear which one the question refers to. In such cases, enumerating all possible candidates would be inefficient, and the most appropriate strategy is to request clarification.

## 3.2 DATASET GENERATION

We construct our dataset using images from the COCO dataset (Lin et al., 2014) as the visual source. To identify objects and potential sources of ambiguity, we rely on the bounding box annotations provided in COCO. These annotations specify the location and category of each object in the scene, enabling a systematic identification of potential ambiguity sources. In particular, bounding boxes allow us to quantify both the number and the spatial prominence of objects, providing a principled basis for controlling ambiguity levels.

For Level 0, we use randomly sampled images and design unambiguous questions such that the target object is explicitly specified without vague referential terms (e.g., "this", "that", "these"). This guarantees a unique, distraction-free interpretation, corresponding to the zero-ambiguity setting. For Level 1, we select images that contain a single salient object. Specifically, Object saliency is determined using a structured scoring scheme. For each object, we compute a saliency score as a weighted combination of its area ratio and its normalized distance to the image center, with weights of 0.7 and 0.3, respectively. The object with the highest score is regarded as the primary candidate. Based on our empirical analysis, objects with a saliency score greater than 0.6 are considered salient. An image is assigned to Level 1 if exactly one object satisfies this criterion. While other minor objects may be present, their visual prominence is negligible, ensuring that vague referential terms can be resolved unambiguously through context. For Level 2, we identify images with a small number of salient objects (two to three bounding boxes above the threshold), where multiple plausible answers exist and enumerating alternatives is the most natural strategy. For Level 3, we target complex scenes with a larger number of salient objects (five or more bounding boxes, often of similar categories or sizes), where ambiguous questions genuinely require explicit clarification.

To generate question–answer pairs for collected images, we employ GPT-5 with level-specific prompts aligned to the above definitions. This controlled prompting procedure ensures that the linguistic form of the questions and the corresponding answer strategies consistently reflect the intended ambiguity level. Please see Appendix K for all prompts used in dataset construction.

## 3.3 DATASET FILTERING

As AQUA is a synthetic dataset, we apply a rigorous multi-stage filtering process to ensure its quality. For this, we adopt a three-stage filtering pipeline: (i) we first verify that each instance satisfies the requirements of its designated ambiguity type; (ii) we then verify if each image–question pair better fits a different ambiguity level, ensuring that the assigned level is uniquely justified by the visual context; and (iii) we confirm that the image is a real-world photograph and validate both the clarity of the question and the factual correctness of the answer. All three stages are evaluated

using GPT-5-mini, and only image–question–answer triplets that pass all stages are retained. Please refer to Appendix K for the dataset filtering prompts.

Through this process, we collect 7.2K samples in total: 3.6K for training and 3.6K for evaluation. Each split is evenly balanced across the four ambiguity levels, with 0.9K instances per level. Please see Appendix A for additional examples of the AQUA.

To ensure the reliability of the evaluation split, we perform human validation on all samples in this split using Amazon Mechanical Turk (MTurk).[4] For each generated sample, annotators verify whether the image–question–answer triplet conforms to its assigned ambiguity level, providing a binary PASS/FAIL judgment. Each instance is independently evaluated by two annotators, and only samples that receive a PASS label from both annotators are retained. Further details of the filtering procedure and annotation protocol are also provided in Appendix B.

### 3.4 HUMAN EVALUATION OF STRATEGIC RESPONSE SELECTION

To examine whether the four strategic response levels defined in AQUA align with how humans handle ambiguity in practice, we conduct a human evaluation on Amazon MTurk. Annotators are provided with concise definitions of three response strategies: (1) answering directly when the target is obvious or inferable from context, (2) listing all plausible targets when their number is small, and (3) requesting clarification when the scene is highly ambiguous, corresponding to Level 0–1, Level 2, and Level 3, respectively.

For each image–question pair, we collect strategy annotations from three independent annotators. Final human labels are assigned via majority vote, and examples without agreement are treated as disagreements. The evaluation set consists of 200 test examples, randomly sampled from the test split with 50 examples from each ambiguity level. Table 1 reports the human evaluation results. Level 0 and 1 achieve near-perfect agreement, indicating strong alignment between human judgments and the strategic response levels defined in AQUA.

Table 1: Evaluation results on human strategic choices.

| Level | Agreement | Disagreement |
|---------|-----------|--------------|
| Level 0 | 50 | 0 |
| Level 1 | 48 | 2 |
| Level 2 | 32 | 18 |
| Level 3 | 32 | 18 |

For Level 2 and 3, agreement remains high overall, although some disagreement arises in highly ambiguous cases, reflecting the inherent subjectivity of these scenarios. A closer examination identifies two common sources of disagreement. First, in borderline cases involving three or four plausible targets, annotators may differ on whether it is preferable to enumerate all options or to request clarification. Second, disagreements can arise from differing judgments of object saliency, where annotators vary in whether they infer a salient target based on visual prominence or focus cues. Notably, such disagreements are rare in Level 1 cases, where saliency is clearly defined. Overall, these results demonstrate that human responses largely align with the strategic response levels defined in AQUA, while highlighting that the boundary between listing multiple possibilities and requesting clarification can be subjective in edge cases.

## 4 EXPERIMENTS

We evaluate a range of open-source and closed-source models on our AQUA to assess their ability to handle ambiguity. In addition, we fine-tune two open-source models to investigate whether VLMs are capable of demonstrating strategic ambiguity-handling.

### 4.1 MODEL TRAINING

To investigate whether VLMs can develop strategic capabilities for handling different types and degrees of ambiguity, we fine-tune Qwen2.5-VL-3B-Instruct and InternVL3-2B-Instruct on AQUA. These models were chosen because (1) they are widely adopted and well-regarded in the research

---

[4] https://www.mturk.com/

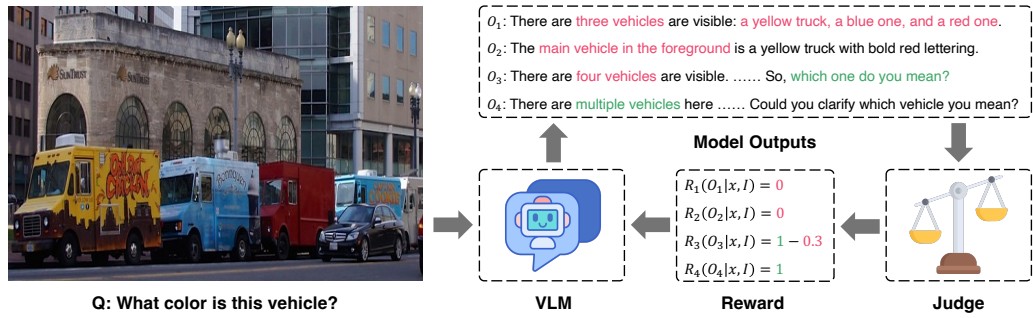

Figure 3: Reward assignment process. Since the given image contains multiple vehicles, the correct response is to request clarification. A perfectly accurate clarification receives a reward of 1. If clarification is requested but contains factual error, a 0.3 penalty is applied. All other response types are assigned a reward of 0.

community, (2) they perform strongly on standard VQA benchmarks, and (3) their parameter sizes offer practical trade-offs between computational efficiency and performance.

**Training Strategy.** We train all models using a two-stage pipeline consisting of supervised fine-tuning (SFT) followed by Group Relative Policy Optimization (GRPO) (Shao et al., 2024). SFT alone does not reliably enforce the correct choice of strategy under different ambiguity levels. To address this limitation, we then apply GRPO, which provides explicit rewards for strategy-aware outputs and thereby strengthens the model's ability to make contextually appropriate decisions.

**Reward Design.** GRPO is conducted under an LLM-as-a-judge framework, where GPT-5-mini serves as the judge (see Appendix K for prompt). For a generated response $y$ given input $(x, I)$, where $x$ denotes the question and $I$ the corresponding image, the reward $R(y|x, I)$ is defined as (Figure 3):

$$R(y|x, I) = \begin{cases} 1 - \lambda & \text{if strategy is correct but factual distortion detected,} \\ 1 & \text{if strategy is correct and no distortion,} \\ 0 & \text{otherwise,} \end{cases}$$

where $\lambda$ denotes the penalty applied if hallucination or factual inconsistency is detected, and is set to 0.3 in our experiments.

**Data Splits.** For SFT, we use the training split of AQUA, dividing it into 80% for training and 20% for validation, ensuring balanced coverage of all four ambiguity levels. For GRPO, we randomly sample 15 instances per level for training and 5 per level for validation from the same split, again maintaining balanced label distribution. Additional optimization details and hyperparameters are provided in Appendix C.

## 4.2 Evaluation Metrics

Our evaluation is performed under an LLM-as-a-judge framework, where GPT-5-mini serves as the judge. To verify the reliability of this automatic evaluation, we sample 400 cases from the test split and compare GPT-5-mini's judgments against an in-house human evaluation, confirming that the automated judgments are highly aligned with human assessment (98.5% agreement). Detailed explanations are provided in Appendix D.

We report two complementary metrics. First, *factual consistency* indicates that the model's response is faithful to the content of the given image, even if not all details are included, and is judged in a binary manner (Grounded or Ungrounded). Second, *strategic accuracy* measures whether the response strategy matches the ground-truth ambiguity level. If a response cannot be reliably mapped to any of the four defined levels, it is assigned an *Unknown* label. This metric is computed over all responses independent of their factual consistency, since our goal is to evaluate the model's ability to choose the correct strategy rather than to remain factually accurate.

Table 2: Main benchmarking results of various VLMs on AQUA. Unk denotes Unknown.

| Model | Factual Acc. | | Strategic Acc. | | | | | |
|---|---|---|---|---|---|---|---|---|
| | Grounded | Ungrounded | Level 0 | Level 1 | Level 2 | Level 3 | Overall | Unk |
| **Zero-shot** | | | | | | | | |
| Qwen2.5-VL-3B-Instruct | 79.86 | 20.14 | 97.11 | 0.11 | 33.33 | 0.78 | 32.83 | 104 |
| Qwen2.5-VL-72B-Instruct | 89.33 | 10.67 | 99.56 | 0.56 | 2.11 | 0.89 | 25.78 | 12 |
| InternVL3-2B-Instruct | 76.63 | 23.37 | 96.0 | 2.33 | 3.56 | 1.89 | 25.95 | 138 |
| InternVL3-78B-Instruct | 80.5 | 19.5 | 96.0 | 2.11 | 3.0 | 5.67 | 26.7 | 133 |
| GPT-5 | 98.4 | 1.6 | 89.67 | 0.67 | 0.33 | 0.78 | 22.86 | 178 |
| Gemini 2.5 Flash | 91.89 | 8.11 | 99.00 | 5.22 | 4.44 | 0.89 | 27.39 | 9 |
| **Chain-of-Thought (CoT)** | | | | | | | | |
| Qwen2.5-VL-3B-Instruct | 78.22 | 21.78 | 95.89 | 8.33 | 5.67 | 3.78 | 28.42 | 60 |
| Qwen2.5-VL-72B-Instruct | 86.97 | 13.03 | 93.0 | 13.78 | 2.78 | 1.33 | 27.72 | 10 |
| InternVL3-2B-Instruct | 76.08 | 23.92 | 97.67 | 2.44 | 1.33 | 1.11 | 25.64 | 54 |
| InternVL3-78B-Instruct | 79.75 | 20.25 | 96.78 | 5.22 | 3.67 | 12.33 | 29.5 | 74 |
| GPT-5 | 98.83 | 1.17 | 97.33 | 3.78 | 0.67 | 1.11 | 25.72 | 14 |
| Gemini 2.5 Flash | 91.64 | 8.36 | 98.0 | 7.89 | 3.56 | 0.22 | 27.42 | 22 |
| **Strategy Prompting** | | | | | | | | |
| Qwen2.5-VL-3B-Instruct | 88.08 | 11.92 | 99.78 | 0.22 | 0.22 | 1.44 | 25.42 | 8 |
| Qwen2.5-VL-72B-Instruct | 91.5 | 8.5 | 99.78 | 5.89 | 17.11 | 46.11 | 42.22 | 12 |
| InternVL3-2B-Instruct | 68.42 | 31.58 | 93.33 | 1.22 | 4.0 | 10.11 | 27.17 | 152 |
| InternVL3-78B-Instruct | 86.44 | 13.56 | 96.89 | 5.56 | 5.89 | 14.11 | 30.61 | 64 |
| GPT-5 | 99.17 | 0.83 | 94.56 | 59.0 | 10.67 | 4.78 | 42.25 | 19 |
| Gemini 2.5 Flash | 94.08 | 5.92 | 99.11 | 8.0 | 10.68 | 30.11 | 36.98 | 35 |
| **AQUA Tuned Models** | | | | | | | | |
| Qwen2.5-VL-3B-Tuned | 81.06 | 18.94 | 99.56 | 77.0 | 82.22 | 86.33 | 86.28 | 1 |
| InternVL3-2B-Tuned | 80.44 | 19.56 | 98.78 | 80.0 | 59.67 | 78.0 | 79.11 | 12 |

# 5 RESULTS

Table 2 shows the performance of a range of VLMs on AQUA. Across all models, factual consistency remains quite high, indicating that hallucinations are rare. The primary challenge, however, lies in strategic reasoning, where performance is poor across all levels except Level 0. This suggests that differences in performance primarily reflect the models' inability to select appropriate ambiguity-handling strategies. Please refer to Appendix E for full benchmarking results, including models of other sizes.

**Base VLMs.** Both open-source models (Qwen2.5-VL-Instruct and InternVL3-Instruct series) and strong closed-source models (GPT-5 and Gemini 2.5 Flash) exhibit similar performance patterns. While these models perform well on unambiguous cases (Level 0), they consistently struggle with ambiguous scenarios (Levels 1–3), showing poor performance when multiple plausible interpretations or clarification requests are required. Notably, even the strongest closed-source models struggle with higher levels of ambiguity, indicating that model scale alone does not resolve the strategic reasoning challenges posed by our dataset (see Appendix F for more details). The same holds for large open-source variants (e.g., Qwen2.5-VL-72B-Instruct and InternVL3-78B-Instruct), which also fail to consistently outperform their smaller counterparts despite their increased size.

**CoT and Strategy-Prompting.** We next examine whether standard prompting techniques improve ambiguity handling. We consider two prompting variants: (i) *Chain-of-Thought (CoT) (Wei et al., 2022)*, where we append *"Let's think step by step."* to encourage stepwise reasoning, and (ii) *Strategy Prompting*, which explicitly instructs the model to choose among four response strategies depending on the level of ambiguity (see Appendix K for prompt). As shown in Table 2, CoT provides no meaningful benefit and often reduces performance, since models tend to generate verbose single-answer responses instead of adapting their strategy. Strategy prompting has no effect on smaller open-source models, but yields slight improvements for larger or stronger closed-source models. These findings suggest that models cannot handle ambiguity reliably through prompting alone and instead need explicit training on datasets like AQUA to acquire strategy-aware response abilities.

**Trained Models.** In contrast, Qwen2.5-VL-3B-Tuned and InternVL3-2B-Tuned models reach approximately 80% overall strategic accuracy, substantially higher than all baselines and prompting-based variants. Importantly, these models maintain robust strategic reasoning across all ambiguity levels. Unlike base VLMs, which default to overconfident single answers, the tuned models reliably

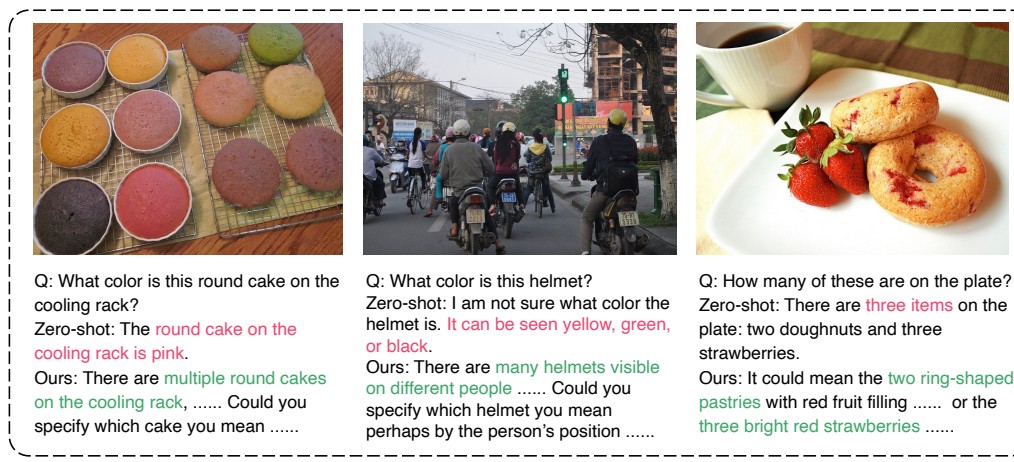

Figure 4: Response comparison of Qwen2.5-VL-3B-Instruct in zero-shot and tuned settings.

Table 3: Performance comparison of models tuned on AQuA with SFT and SFT+GRPO. G, U, and Unk respectively denote Grounded, Ungrounded, and Unknown.

| Model | Factual Acc. | | Strategic Acc. | | | | | |
|---|---|---|---|---|---|---|---|---|
| | G | U | Level 0 | Level 1 | Level 2 | Level 3 | Overall | Unk |
| Qwen2.5-VL-3B-Tuned (SFT) | 82.78 | 17.22 | 99.56 | 92.22 | 61.33 | 82.11 | 83.81 | 2 |
| Qwen2.5-VL-3B-Tuned (SFT+GRPO) | 81.06 | 18.94 | 99.56 | 77.0 | 82.22 | 86.33 | 86.28 | 1 |
| InternVL3-2B-Tuned (SFT) | 66.08 | 33.92 | 99.22 | 82.67 | 37.67 | 74.11 | 73.42 | 2 |
| InternVL3-2B-Tuned (SFT+GRPO) | 80.44 | 19.56 | 98.78 | 80.0 | 59.67 | 78.0 | 79.11 | 12 |

adapt their strategies. This consistent behavior shows that explicit training on AQuA enables models to handle visual ambiguity in a human-like and strategy-aware manner. Please refer to Figure 4 for examples of our model's strategic response.

## 6 ANALYSIS

### 6.1 SFT AND RL TRAINING

To better understand the effect of each training stage, we conduct an ablation comparing models trained with SFT alone against those further optimized with GRPO. As shown in Table 3, models trained with SFT alone already achieve over 73% strategic accuracy overall, confirming that simple supervised training on ambiguity-aware responses is sufficient to yield strong gains. Nonetheless, performance on highly ambiguous cases (Levels 2 and 3) remains lower. Applying GRPO further boosts performance, this stage not only raises accuracy on Levels 2 and 3, but also stabilizes performance more broadly, leading to balanced and robust strategic reasoning. However, we observe a slight drop in Level 1 performance after applying GRPO following SFT (Figure 5b and 5c). We find that models trained only with SFT tend to concentrate most of their errors in Level 1, indicating either a form of overfitting to that level or an insufficient understanding of Levels 2 and 3. As GRPO encourages more strategic decision-making across all ambiguity levels, this bias is mitigated, and the resulting redistribution of errors naturally leads to a minor decrease in Level 1 accuracy. Please see Appendix G for confusion matrices of InternVL3-2B based models. Also, please refer to Appendix H for additional results with increased GRPO training samples.

### 6.2 ERROR PATTERNS

**Biased Default Strategy of VLMs.** Figure 5 presents the confusion matrices of Qwen2.5-VL-3B-Instruct and Qwen2.5-VL-3B-Tuned (SFT+GRPO) evaluated on AQuA. In the base model (Figure 5a), we observe a strong bias toward Level 0 predictions, where the model outputs a single confident answer even when ambiguity requires context inference (Level 1), multiple listings (Level 2), or explicit clarification (Level 3). This indicates that the model defaults to a *one-correct-answer*

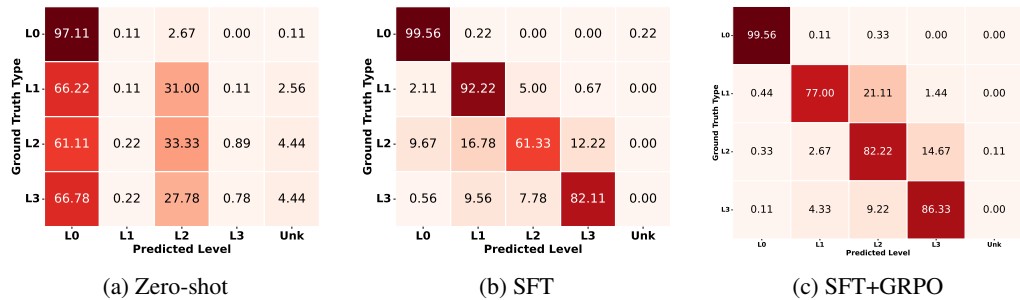

(a) Zero-shot      (b) SFT      (c) SFT+GRPO

Figure 5: Confusion matrices of the response patterns of Qwen2.5-VL-3B-Instruct on the AQUA.

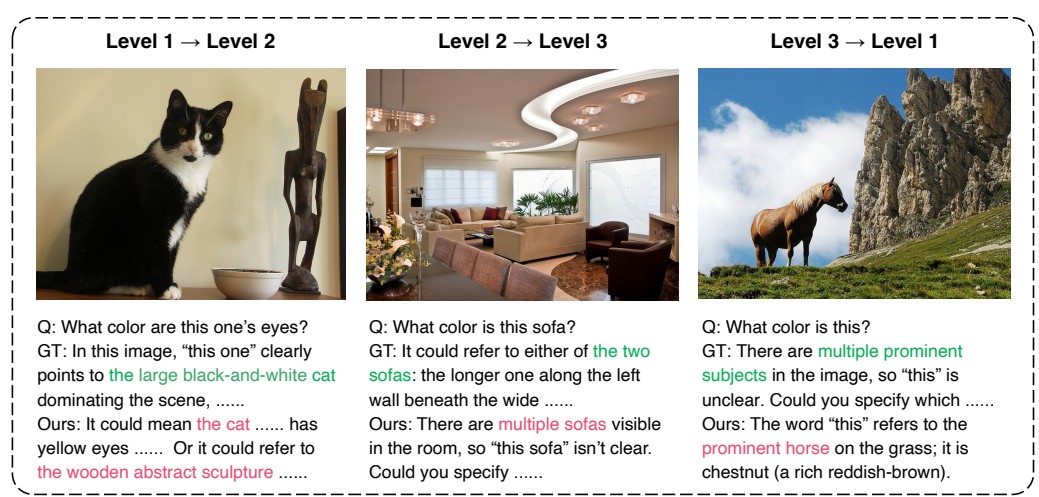

Figure 6: Our model responses to level-boundary confusion and salience-driven errors.

strategy regardless of the degree of ambiguity. Similar patterns are observed in other baseline models. However, Qwen2.5-VL-3B-Instruct shows an unusually high proportion of Level 1 predictions.[5]

**Confusion at Level Boundaries.** As shown in Figure 5b, the SFT model tends to collapse toward Level 1 responses, rather than selecting appropriate response strategies. Many error cases are misclassified as Level 1, suggesting that the model overfits to a specific strategy instead of accurately reasoning about the underlying ambiguity. In contrast, after fine-tuning with SFT+GRPO (Figure 5c), the model shows substantial improvements across all ambiguity levels. Notably, the strong bias toward Level 1 is greatly reduced, resulting in predictions that are more evenly distributed across the intended strategies. However, occasional confusions remain near the boundaries between levels, often driven by stereotypes or conventional expectations. As shown in Figure 6, the left example is labeled as Level 1 because humans typically associate it with the cat, however our model also considers the sculpture's "eyes" shifting its response to Level 2. Also, the middle example, our model treats them as distinct objects rather than a single set and requests clarification. These examples highlight how subtle biases and interpretation choices can shift predictions across adjacent ambiguity levels.

**Salience-Driven Errors.** Although uncommon, the model occasionally deviates from the appropriate strategy by focusing on a salient feature unintended by the query. In the right example of Figure 6, the question *"What color is this?"* is paired with an image containing the sky, rocks, grass, and a horse. The ground-truth response requires a clarification request because multiple plausible referents exist. However, our model interprets the horse as the intended referent, as it is the most salient object in the scene, and directly answers with its color. This results in a shift from Level 3 to Level 1. Such cases often arise from salient or stereotypical features that lead the model to overcommit to a single referent instead of requesting clarification or listing alternatives.

---

[5]Qwen2.5-VL-3B-Instruct tends to answer with "I am not sure ... It could be *A*, *B*, or *C* ..." when it cannot make a clear decision, regardless of the ambiguity level.

## 6.3 Effectiveness of Clarification

In cases of high ambiguity, the model tends to ask for clarification. To assess the effectiveness of this strategy, we design an experiment in a two-turn question–answer setting. Specifically, we filter 100 Level 3 instances and use GPT-5 to generate a follow-up turn consisting of a single disambiguating hint and the corresponding unambiguous answer (see Appendix I for examples).

For each response, GPT-5-mini serves as the judge, assigning a binary PASS or FAIL depending on whether the model's final answer matches the ground-truth unambiguous answer (see Appendix K for prompt). As summarized in Table 4, both models achieve consistently high PASS rates, once a clarifying hint is provided, demonstrating that Level 3 ambiguity can be effectively resolved with a single clarification turn. These findings highlight the value of clarification:

Table 4: Evaluation results on the clarification subset.

| Model | PASS | FAIL |
|---|---|---|
| Qwen2.5-VL-3B-Tuned | 80% | 20% |
| InternVL3-2B-Tuned | 73% | 27% |

with a short follow-up, the model can resolve uncertainty and provide accurate, well-grounded answers rather than enumerating all possible answers in the first place.

Table 5: Evaluation results of samples generated using Open Images V7.

| Model | Level 0 | Level 1 | Level 2 | Level 3 | Overall | Unk |
|---|---|---|---|---|---|---|
| Qwen2.5-VL-3B-Instruct | 98.0 | 2.0 | 1.0 | 0 | 25.25 | 2 |
| Qwen2.5-VL-3B-Tuned | 97.0 | 87.0 | 76.0 | 89.0 | 83.81 | 2 |
| InternVL3-2B-Instruct | 98.0 | 1.0 | 5.0 | 1.0 | 26.25 | 7 |
| InternVL3-2B-Tuned | 96.0 | 86.0 | 55.0 | 77.0 | 78.5 | 2 |

## 6.4 Generalization Beyond COCO

AQUA is constructed using images from the COCO dataset, and our model is trained on the same source. To evaluate generalization beyond COCO, we test the model on ambiguous image–question pairs from a different image dataset. Using the same generation procedure (including human filtering), we construct an evaluation set from Open Images V7[6] with 100 samples per ambiguity level (400 samples in total). As shown in Table 5, performance on Open Images follows trends similar to those observed on COCO-based samples. Although trained only on COCO images, our models consistently select appropriate strategies across all ambiguity levels. This suggests that the models generalize beyond dataset-specific visual characteristics. Overall, these results indicate that our approach is robust to dataset shift and can handle ambiguity in VQA across different image sources.

## 7 Conclusion

In this work, we introduce AQUA, a new dataset designed not only to evaluate but also to train VLMs in handling ambiguity in VQA. AQUA defines four fine-grained levels, each aligned with a distinct response strategy. Through this design, we show that current VLMs often fail to recognize and adapt to different types of ambiguity, defaulting to overconfident answers rather than reasoning strategically. By fine-tuning open-source models with supervised learning and GRPO on AQUA, we demonstrate that even relatively small VLMs can learn to choose strategies contextually—whether by direct answering, inference from context, controlled enumeration, or explicit clarification. These tuned models outperform both larger open-source and strong closed-source systems on ambiguous VQA, highlighting the effectiveness of strategy-aware training. In addition, we conduct an extensive analysis to understand why VLMs fail to generate strategy-aware responses under ambiguity. Untuned models often do not even recognize when a question–image pair is ambiguous, leading them to produce overconfident answers. In contrast, failures in our tuned models mostly arise in boundary cases, where ambiguity levels are difficult to distinguish, or from salience-driven errors, where prominent visual features bias the response. These findings provide a deeper explanation of the limitations of current VLMs and point toward the need for models that can reason more flexibly about uncertainty.

---

[6]https://storage.googleapis.com/openimages/web/index.html

## ACKNOWLEDGMENTS

We thank the reviewers and the area chair for their valuable feedback. This work was partly supported by Institute of Information & communications Technology Planning & Evaluation (IITP) grant funded by the Korea government(MSIT) (No.RS-2019-II191906, Artificial Intelligence Graduate School Program(POSTECH)), and National Research Foundation of Korea(NRF) grant funded by the Korea government(MSIT)(RS-2025-23612977), and Basic Science Research Program through the National Research Foundation of Korea(NRF) funded by the Ministry of Education(RS-2025-25433785).

## REPRODUCIBILITY STATEMENT

We provide samples of the AQUA and the training code in the supplementary materials. After the review process is complete, we will publicly release the full dataset, model checkpoints, and all source code to ensure reproducibility. In addition, implementation details for training are described in Section 4.1 and Appendix C, and all prompts used in this study are provided in Appendix K and can be used to fully reproduce our experiments.

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

## A  EXAMPLES OF AQUA

Please refer to Figure 7 for Level 0, Figure 8 for Level 1, Figure 9 for Level 2, and Figure 10 for Level 3 of the AQUA dataset.

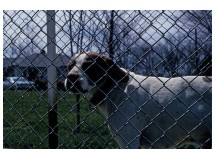

Q: What animal is shown behind the chain-link fence?
A: The animal behind the chain-link fence is a dog.

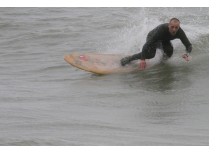

Q: What water sport is the person wearing a black wetsuit performing?
A: The person is surfing.

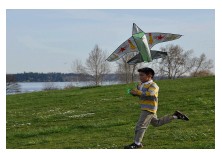

Q: what is the kite shaped like?
A: The kite is shaped like an airplane.

Figure 7: Level 0 examples of AQUA.

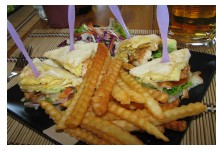

Q: What style are these cut in?
A: The word "these" refers to the big pile of French fries in the foreground, and they are crinkle-cut.

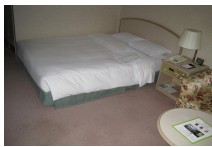

Q: How many pillows are on it?
A: In this image, "it" refers to the large bed dominating the scene, and there are two pillows on it.

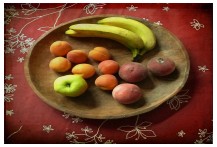

Q: What is this made of?
A: The word "this" points to the large round container that dominates the scene-the bowl holding the fruit-and it is made of wood.

Figure 8: Level 1 examples of AQUA.

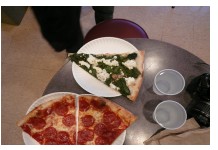

Q: What is on this plate?
A: It could refer to the upper plate, which holds one slice of pizza topped with leafy spinach and white cheese. Or it could refer to the lower plate, which holds two slices of pepperoni pizza with red pepperoni discs.

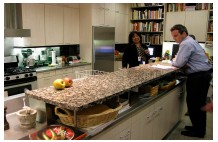

Q: What color are these apples?
A: If you mean the two apples on the raised granite countertop near the center, one is red, and the other is greenish-yellow.

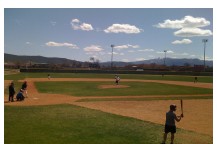

Q: Where is this person with a bat standing?
A: One interpretation is the batter at home plate on the left, wearing a red helmet and standing in the batter's box beside the catcher and umpire. Another interpretation is the on-deck batter in the right foreground, holding a bat while standing on the dirt near the foul line at the edge of the grass.

Figure 9: Level 2 examples of AQUA.

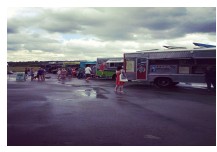

Q: What color is this food truck?
A: There are multiple food trucks visible, so "this" could refer to more than one. Could you specify which truck you mean (for example, by its position or a detail on it)? That clarification is needed to identify the correct target.

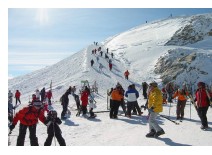

Q: What color is this skier's jacket?
A: There are many skiers visible, so I'm not sure which one you mean. Could you specify which skier you're referring to (for example, their position in the image)? I need clarification because multiple jackets are equally prominent and "this skier" isn't uniquely identifiable.

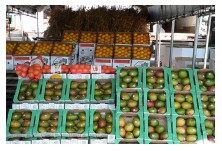

Q: How many fruits are in this box?
A: There are several boxes visible, so I'm not sure which one you mean. Could you specify the box by its position or a nearby detail? I need clarification because multiple boxes are equally prominent and 'this box' could refer to any of them.

Figure 10: Level 3 examples of AQUA.

**Instructions**

1. Look at the image.
2. Read the question and the given answer.
3. Check the provided **Ambiguity Level definition** and compare with the example cases:

**Please note that a random quality check question may appear. If you receive this question, you must pass it for your work to be approved.**

- **Level 0 – No ambiguous terms**
  *Q:* What color is the object the girl is holding in her raised hand?
  *A:* The object is white.
  ✅ Direct factual answer, single clear interpretation.
- **Level 1 – Ambiguous term (e.g., "this", "that", "it", "the thing"), but one dominant target**
  *Q:* What is this doing with its trunk?
  *A:* In this scene, "this" means the large elephant in the middle, and it is feeding—grabbing and eating the surrounding vegetation with its trunk.
  ✅ Ambiguity resolved by explicitly stating the referent, then giving the answer.
- **Level 2 – Ambiguous with 2–3 plausible targets**
  *Q:* What color are these on top?
  *A:* They could be the two small hippopotamus figurines standing near the center of the cake, which are light gray; or the whole strawberries arranged on the cake, which are bright red with green leaves.
  ✅ All plausible targets are enumerated, no single best guess.
- **Level 3 – Ambiguous with many possible targets**
  *Q:* What color is this on the path?
  *A:* There are multiple visible subjects here, so I'm not sure which one "this" refers to. Could you specify which item you mean? I need that clarification to describe the correct color for the intended target.
  ✅ Too many possible interpretations → politely ask for clarification.

---

**Image**

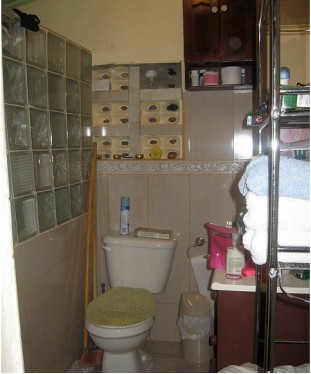

---

**Q&A**

**Question:**     What color is this towel?

**Answer (given):**    I see multiple towels in the image, so I'm not sure which one you mean. Could you specify which towel (for example, its position or what it's next to)? Clarification is needed because "this" doesn't uniquely identify a single object here.

**Level (0–3):**     3

---

**Select your evaluation:**

◯ PASS ✅     ◯ FAIL ❌

Submit

Figure 11: MTurk Worker Evaluation Screen.

## B  DATASET FILTERING METHOD

To ensure the quality of AQUA, we designed a three-stage filtering pipeline:

- **Stage 1 - Level Consistency Check**: This stage verifies that each generated question–answer pair satisfies the requirements of its assigned ambiguity level. For example, Level 0 samples must contain no ambiguous terms and allow only one definitive answer, while Level 1 samples must contain at least one ambiguous term but resolve it confidently in the answer. This acts as a strict rule-based gate to filter out obvious mislabeling (e.g., a Level 0 example using "this", or a Level 2 answer that selects only one option).

- **Stage 2 - Best Fit Validation**: Even if a sample meets the basic criteria of its assigned level, it may be more appropriately categorized under a different level. This stage checks whether the assigned level is the unique best fit among the four categories. LLM-as-a-judge compares the question–answer pair against canonical definitions and applies explicit priority rules. For example, if a question uses an ambiguous term but only one dominant object is present, Level 1 is always preferred. This ensures that each retained sample is not only valid but also aligned with the most specific ambiguity level.

- **Stage 3 - Real-World and Quality Validation**: The final stage ensures that each sample is suitable for inclusion in a real-world VQA dataset. This includes (i) confirming that the underlying image is a natural photograph with sufficient clarity, (ii) verifying that the question refers only to observable properties (e.g., color, shape, size, count) without requiring hidden knowledge, and (iii) checking that the answer is grounded in the image and consistent with the behavioral expectations of its level. This stage also eliminates degenerate cases such as synthetic or corrupted images, or hallucinated content in answers.

After applying the three-stage filtering process to all data samples, we further enhance the reliability of AQUA by conducting an additional human validation stage for the evaluation split. This step is carried out on the Amazon Mechanical Turk (MTurk) platform, where we restrict participation to workers with more than 5K previously approved HITs and an approval rate above 95%. Annotators are presented with the image, question, and answer, and asked to judge, considering the assigned ambiguity level, whether the sample is acceptable, providing a binary PASS/FAIL decision. Each sample is independently evaluated by two annotators, and only those that receive a PASS label from both are retained in the dataset. As an additional safeguard, we inject 10% fake samples into the annotation pool. If a worker incorrectly assigns a PASS label to any fake sample, all of their submitted annotations are discarded. Please see Figure 11 for the instructions and interface used in the human validation stage.

## C  IMPLEMENTATION DETAILS

Our training procedure consists of two stages: (1) supervised fine-tuning (SFT) and (2) Group Relative Policy Optimization (GRPO). All trainings are conducted on 8 NVIDIA RTX A6000 GPUs.

For SFT, we fully fine-tune Qwen2.5-VL-3B-Instruct using the HuggingFace Trainer with the AdamW optimizer, a learning rate of $5 \times 10^{-5}$, a constant_with_warmup scheduler with a warmup ratio of 0.03, and gradient checkpointing enabled. Training is performed for 3 epochs with an auto-fined per-device batch size and a gradient accumulation step of 4, and gradients are clipped at 1.0. For InternVL3-2B-Instruct, we also fully fine-tune the model using the official InternVL training script with the AdamW optimizer, a learning rate of $2 \times 10^{-5}$, a weight decay of 0.05, a `cosine` learning rate scheduler with a warmup ratio of 0.03, and gradient checkpointing. Training is conducted for 3 epoch with a per-device batch size of 4 and a gradient accumulation step of 4. We apply early stopping with a patience of 1 for both models and select the best-performing checkpoint accordingly.

For GRPO, we adapt the training scripts released by Fan et al. (2025). The reward function is implemented with GPT-5-mini. We train for 30 epochs with a learning rate of $5 \times 10^{-6}$, batch size of 2, gradient accumulation steps of 2, and $\beta = 0.01$, using a cosine learning rate scheduler. For each sample, we generate four responses, compute rewards for each, and update the model using group-based advantages combined with KL divergence against a reference model. We select the final checkpoint based on the highest validation reward.

Table 6: Full benchmarking results of various VLMs on AQuA. G, U, and Unk respectively denote Grounded, Ungrounded, and Unknown.

| Model | Factual Acc. | | Strategic Acc. | | | | | |
|---|---|---|---|---|---|---|---|---|
| | G | U | Level 0 | Level 1 | Level 2 | Level 3 | Overall | Unk |
| **Zero-shot** | | | | | | | | |
| Qwen2.5-VL-3B-Instruct | 79.86 | 20.14 | 97.11 | 0.11 | 33.33 | 0.78 | 32.83 | 104 |
| Qwen2.5-VL-7B-Instruct | 87.97 | 12.03 | 98.78 | 0.78 | 3.67 | 3.33 | 26.64 | 25 |
| Qwen2.5-VL-72B-Instruct | 89.33 | 10.67 | 99.56 | 0.56 | 2.11 | 0.89 | 25.78 | 12 |
| InternVL3-2B-Instruct | 76.63 | 23.37 | 96.0 | 2.33 | 3.56 | 1.89 | 25.95 | 138 |
| InternVL3-8B-Instruct | 81.52 | 18.48 | 97.67 | 1.67 | 2.11 | 2.67 | 26.03 | 94 |
| InternVL3-78B-Instruct | 80.5 | 19.5 | 96.0 | 2.11 | 3.0 | 5.67 | 26.7 | 133 |
| GPT-5 | 98.4 | 1.6 | 89.67 | 0.67 | 0.33 | 0.78 | 22.86 | 178 |
| Gemini 2.5 Flash | 91.89 | 8.11 | 99.00 | 5.22 | 4.44 | 0.89 | 27.39 | 9 |
| **Chain-of-Thought (CoT)** | | | | | | | | |
| Qwen2.5-VL-3B-Instruct | 78.22 | 21.78 | 95.89 | 8.33 | 5.67 | 3.78 | 28.42 | 60 |
| Qwen2.5-VL-7B-Instruct | 83.69 | 16.31 | 88.0 | 11.46 | 5.01 | 2.89 | 26.85 | 31 |
| Qwen2.5-VL-72B-Instruct | 86.97 | 13.03 | 93.0 | 13.78 | 2.78 | 1.33 | 27.72 | 10 |
| InternVL3-2B-Instruct | 76.08 | 23.92 | 97.67 | 2.44 | 1.33 | 1.11 | 25.64 | 54 |
| InternVL3-8B-Instruct | 76.17 | 23.83 | 95.22 | 7.67 | 3.0 | 9.11 | 28.74 | 127 |
| InternVL3-78B-Instruct | 79.75 | 20.25 | 96.78 | 5.22 | 3.67 | 12.33 | 29.5 | 74 |
| GPT-5 | 98.83 | 1.17 | 97.33 | 3.78 | 0.67 | 1.11 | 25.72 | 14 |
| Gemini 2.5 Flash | 91.64 | 8.36 | 98.0 | 7.89 | 3.56 | 0.22 | 27.42 | 22 |
| **Strategy Prompting** | | | | | | | | |
| Qwen2.5-VL-3B-Instruct | 88.08 | 11.92 | 99.78 | 0.22 | 0.22 | 1.44 | 25.42 | 8 |
| Qwen2.5-VL-7B-Instruct | 90.64 | 9.36 | 99.67 | 0.78 | 1.33 | 10.33 | 28.03 | 16 |
| Qwen2.5-VL-72B-Instruct | 91.5 | 8.5 | 99.78 | 5.89 | 17.11 | 46.11 | 42.22 | 12 |
| InternVL3-2B-Instruct | 68.42 | 31.58 | 93.33 | 1.22 | 4.0 | 10.11 | 27.17 | 152 |
| InternVL3-8B-Instruct | 78.03 | 21.97 | 90.67 | 11.11 | 9.67 | 17.11 | 32.14 | 57 |
| InternVL3-78B-Instruct | 86.44 | 13.56 | 96.89 | 5.56 | 5.89 | 14.11 | 30.61 | 64 |
| GPT-5 | 99.17 | 0.83 | 94.56 | 59.0 | 10.67 | 4.78 | 42.25 | 19 |
| Gemini 2.5 Flash | 94.08 | 5.92 | 99.11 | 8.0 | 10.68 | 30.11 | 36.98 | 35 |
| **AQuA Tuned Models** | | | | | | | | |
| Qwen2.5-VL-3B-Tuned (SFT) | 82.78 | 17.22 | 99.56 | 92.22 | 61.33 | 82.11 | 83.81 | 2 |
| Qwen2.5-VL-3B-Tuned (SFT+GRPO) | 81.06 | 18.94 | 99.56 | 77.0 | 82.22 | 86.33 | 86.28 | 1 |
| InternVL3-2B-Tuned (SFT) | 66.08 | 33.92 | 99.22 | 82.67 | 37.67 | 74.11 | 73.42 | 2 |
| InternVL3-2B-Tuned (SFT+GRPO) | 80.44 | 19.56 | 98.78 | 80.0 | 59.67 | 78.0 | 79.11 | 12 |

## D    VERIFICATION FOR LLM-AS-A-JUDGE

To verify the reliability of our LLM-as-a-judge framework, we conduct an in-house evaluation on a sample of responses from Qwen2.5-VL-3B-Instruct and Qwen2.5-VL-3B-Tuned. Specifically, we randomly sample 400 responses: 100 classified as Grounded and 100 classified as Ungrounded for factual consistency, and 50 from each ambiguity level for strategic accuracy. Human annotators then independently assess whether GPT-5-mini's judgments are correct.

The results show a high degree of agreement between GPT-5-mini and human evaluation. Out of the 400 sampled cases, only 5 are misclassified in factual consistency and 1 in strategic accuracy, resulting in an overall agreement of 98.5%. This strong alignment demonstrates that GPT-5-mini serves as a reliable judge for our evaluation protocol and confirms that our automatic evaluation is trustworthy for large-scale benchmarking.

## E    FULL BENCHMARKING RESULTS

Please see Table 6 for full benchmark results for a range of VLMs.

Table 7: Evaluation results of iterative prompting with GPT-5 using repeated judging and correction.

| Stage | Level 0 | Level 1 | Level 2 | Level 3 | Overall | Unk |
|---|---|---|---|---|---|---|
| Stage 1 | 95.0 | 60.0 | 11.0 | 3.0 | 42.25 | 1 |
| Stage 2 | 97.0 | 77.0 | 14.0 | 8.0 | 49.0 | 5 |
| Stage 3 | 98.0 | 83.0 | 19.0 | 15.0 | 53.75 | 3 |
| Stage 4 | 97.0 | 89.0 | 25.0 | 16.0 | 56.75 | 6 |
| Stage 5 | 98.0 | 91.0 | 34.0 | 14.0 | 59.25 | 11 |
| Stage 6 | 99.0 | 91.0 | 35.0 | 16.0 | 60.25 | 7 |
| Stage 7 | 98.0 | 94.0 | 41.0 | 17.0 | 62.5 | 5 |
| Stage 8 | 100.0 | 97.0 | 39.0 | 16.0 | 63.0 | 2 |
| Stage 9 | 100.0 | 96.0 | 39.0 | 20.0 | 63.75 | 7 |
| Stage 10 | 100.0 | 95.0 | 43.0 | 19.0 | 64.25 | 6 |

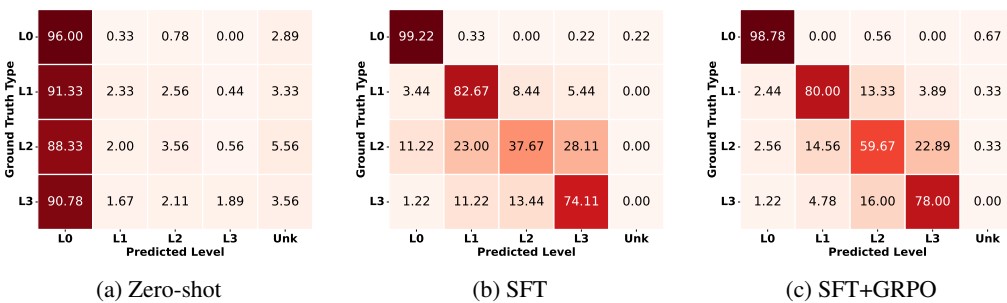

(a) Zero-shot      (b) SFT      (c) SFT+GRPO

Figure 12: Confusion matrices of the response patterns of InternVL3-2B-Instruct on the AQUA.

## F  ITERATIVE PROMPTING ANALYSIS

In our experiments, we observe that even the strongest closed-source models struggle with strategic handling of visual ambiguity. To examine whether improved prompting or more explicit guidance can help such models better resolve ambiguity, we also evaluate an iterative prompting setup. In this setup, GPT-5 repeatedly generates, judges, and revises its responses based on predefined strategy guidelines. We conduct experiments on 200 test examples (50 per ambiguity level) over 10 refinement iterations.

As shown in Table 7, while iterative prompting leads to modest improvements, it remains ineffective on highly ambiguous cases and incurs substantial computational and latency overhead. Overall, this approach fails to achieve the level of consistency observed in the fine-tuned model, suggesting that fine-tuning is necessary for robust, level-aware strategic responses to ambiguity.

## G  ANALYSIS OF ERROR PATTERNS

The confusion matrices of InternVL3-2B-Instruct and InternVL3-2B-Tuned (SFT+GRPO) on AQUA are shown in Figure 12.

## H  INCREASING GRPO TRAINING SAMPLES

After SFT, we train the model using GRPO with 60 samples (15 examples per level). To examine how performance depends on the number of GRPO training samples, we conduct an additional experiment using 120 samples (30 examples per level). As shown in Table 8, increasing the number of training samples from 60 to 120 leads to improved performance on strategic choice.

Table 8: Evaluation results with different numbers of GRPO training samples. In the model names, "60" and "120" indicate models trained with 60 and 120 GRPO samples, respectively.

| Model | Level 0 | Level 1 | Level 2 | Level 3 | Overall | Unk |
|---|---|---|---|---|---|---|
| Qwen2.5-VL-3B-Tuned-60 | 99.56 | 77.0 | 82.22 | 86.33 | 86.28 | 1 |
| Qwen2.5-VL-3B-Tuned-120 | 99.44 | 86.44 | 73.33 | 92.11 | 87.83 | 2 |
| InternVL3-2B-Tuned-60 | 98.78 | 80.0 | 59.67 | 78.0 | 79.11 | 12 |
| InternVL3-2B-Tuned-120 | 96.0 | 90.3 | 57.89 | 85.33 | 82.39 | 29 |

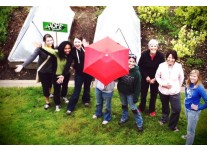
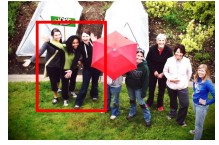

$Q_{ambig}$: How many people are standing to the left of this person?

$A_{clarify}$: I'm not sure which person you're referring to—there are several people visible. Could you specify which one you mean so I can count accurately? Clarification is necessary because the number changes depending on the chosen person.

$Q_{resolved}$: Count the number of people standing to the left of the person holding the red umbrella in the center.

$A_{gold}$: There are three people standing to the left of the person with the red umbrella.

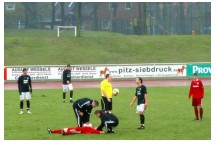
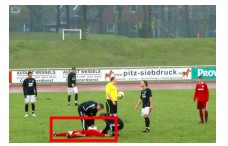

$Q_{ambig}$: What color is this player's jersey?

$A_{clarify}$: There are several players visible, so I can't tell which person you mean. Which player are you referring to? I need clarification because "this player" could be any of them and the image doesn't indicate a specific target.

$Q_{resolved}$: The reference is to the jersey color of the player lying on the ground in the foreground being attended by two men in black.

$A_{gold}$: The player's jersey is red.

Figure 13: Examples from Level 3 of the AQUA dataset, where follow-up questions are used to resolve ambiguity and arrive at the gold answer.

These results suggest that increasing the number of GRPO training samples can further benefit performance. In this work, we strategically adopt 60 samples to balance training time and computational cost. Since the goal of this work is not to identify the optimal sample size, we do not explore smaller or substantially larger sample settings. We leave a more systematic investigation of the effect of GRPO sample size to future work.

## I EXAMPLES OF CLARIFICATION SUBSET FOR LEVEL 3

Please refer to Figure 13 for Level 3 subset examples, where follow-up questions help resolve ambiguity and guide the model to the intended gold answer.

## J THE USE OF LARGE LANGUAGE MODELS

We use large language models (LLMs) throughout this work to support tasks such as data generation, filtering, and reward modeling. During these processes, the authors remain actively involved, carefully reviewing the process and outputs to ensure that results are not solely produced by LLMs but refined through human review and curation. We also use LLMs during the paper writing stage for grammar checking and minor language refinement. However, the overall writing of the paper are conducted directly by the authors without relying on LLMs.

## K    PROMPTS LIST

---

**Prompt for Level 0 Sample Generation**

### Instruction:

Given an image, create exactly one question–answer pair that is clear, factual, and unambiguous, such that only one correct answer exists.

### Requirements:

Must:

- Ensure the question has a single, definitive interpretation.

- Base the question entirely on factual, observable elements in the image.

- Phrase both the question and the answer in complete, clear sentences.

- Ensure the answer is definitively determinable from the image without external knowledge.

Avoid:

- Any ambiguous terms like "this", "that", "it", or "there".

- Subjective or interpretive elements (e.g., opinions, feelings, aesthetic judgments).

- Questions that allow multiple plausible answers.

### Output Format:

{Question: [Your generated question], Answer: [The definitive answer]}

If the image does not allow for such a question–answer pair, output exactly: none

Do not include explanations or additional text.

### Response:

---

**Prompt for Level 1 Sample Generation**

### Instruction:

Given an image, create exactly one question–answer pair where:

- The question uses an ambiguous term, but the image context makes the intended subject completely clear.

- The answer must explicitly resolve the ambiguity first and then give the factual answer.

### Must:

- Use at least one ambiguous term (e.g., "this", "that", "it", "the thing", etc.).

- Ensure there is exactly one clearly dominant object in the image that stands out from all others in size, position, or salience.

- In the answer, naturally explain what the ambiguous term refers to in this specific image, then provide the definitive descriptive answer.

- Write the answer in full sentences.

### Avoid:

- Questions that would remain clear without ambiguous terms.

- Scenes with multiple objects of equal prominence.

- Answers that only give the fact without clarifying the referent.

- Overly short or one-word answers.

---

- Beginning with fixed templates such as "Here, 'this' refers to ...". Each answer must be phrased naturally and vary across samples.

### Output Format:

{Question: [Your ambiguous question], Answer: [Your natural clarification plus the definitive descriptive answer]}

If the image does not meet requirements, output exactly: none

Do not include explanations or any extra text.

### Response:

---

**Prompt for Level 2 Sample Generation**

### Instruction:

Given an image, create exactly one question–answer pair where:

- The question is ambiguous and could reasonably refer to multiple distinct objects in the scene.

- The answer lists all plausible interpretations rather than choosing only one.

### Must:

- Ensure the image contains at least two and at most three reasonable target objects.

- Use ambiguous terms (e.g., "this", "that", "they", "these", etc.) in the question.

- Clearly describe each possible target in the answer so that the user can decide.

- Make each description factual and directly observable from the image.

### Avoid:

- Questions that clearly refer to only one object.

- Scenes where one object completely dominates.

- Scenes with more than three equally plausible candidates.

- Random guessing or adding details not visible in the image.

### Output Format:

{Question: [Your ambiguous question], Answer: [Natural, descriptive sentences listing each plausible interpretation]}

If the image does not meet requirements, output exactly: none

Do not include explanations or any extra text.

### Response:

---

**Prompt for Level 3 Sample Generation**

### Instruction:

Given an image, create exactly one question–answer pair where:

- The question contains ambiguous references, and the image provides no clear clues to identify the intended target.

- The answer requests clarification without attempting to guess or list possible options.

### Must:

- Include at least one ambiguous term (e.g., "this", "that", "it", "the thing", etc.).

- Ensure there are multiple equally prominent objects in the image.

- Make the question about clearly visible, observable properties (e.g., color, shape, size, position, visible text, count, material, etc.).

- In the answer, acknowledge that there are multiple possible targets and politely ask which one is intended.

- Briefly explain why clarification is necessary.

### Avoid:

- Listing all possible targets.

- Making any guesses or inferences.

- Using scenes where one object is clearly dominant.

- Asking about non-observable or speculative properties (e.g., device state, hidden contents, functionality, temperature, brand unless clearly visible).

### Output Format:

{Question: [Your ambiguous question], Answer: [Your clarification request]}

If the image does not meet requirements, output exactly: none

Do not include explanations or any extra text.

### Response:

---

**Prompt for Filtering Stage 1**

### Role You are the LEVEL CHECKER. Decide ONLY whether the given (Level, Question, Answer) correctly demonstrates the REQUIRED behavior for the assigned Level.

If any requirement is not satisfied, output FAIL. Do NOT suggest other Levels or reassign.

### Level Requirements (must ALL hold)

Level 0 (Clear VQA): - Question: clear, specific, and unambiguous; no demonstratives like "this/that/it".

- Answer: direct, factual, determinable from the image; full sentence allowed.

- PASS if: no ambiguous references and only one definitive interpretation.

- FAIL if: ambiguous terms appear OR multiple interpretations exist.

Level 1 (Context-resolved ambiguity):

- Question: contains at least one ambiguous term (e.g., "this", "that", "it", "the thing").

- Context: exactly one clearly dominant target makes the reference obvious.

- Answer: must explicitly clarify what the ambiguous term refers to, and then provide the factual description in a natural full sentence.

- PASS if: the answer both (1) resolves the referent of the ambiguous term and (2) provides a definitive, image-grounded answer in natural language.

- FAIL if: no ambiguous term OR multiple objects are equally prominent OR the answer skips the clarification step OR the answer is just a single word/short fragment.

Level 2 (List all plausible options):

- Question: ambiguous with 2–3 plausible targets.

- Answer: enumerates ALL plausible targets (do NOT pick one best guess); each described clearly and factually, phrased in natural sentences rather than bullet points.

- PASS if: 2–3 plausible targets exist and the answer lists them all in natural descriptive sentences.

- FAIL if: one target dominates OR more than three plausible targets OR the answer picks a single guess OR the answer is in bullet/fragmented list style.

Level 3 (Clarification required):

- Question: ambiguous with many or equally plausible targets; no reasonable best guess.

- Answer: politely requests clarification, acknowledges multiple possibilities WITHOUT listing them, and briefly explains why clarification is needed.

- PASS if: inference is not reasonable and the answer requests clarification (no listing, no guessing).

- FAIL if: one object is clearly more prominent OR a reasonable best guess exists OR the answer lists options.

### Universal Quality Checks (all Levels must satisfy):

- Question concerns visible, observable properties (color/shape/size/position/count/visible text/material).

- No speculative/hidden-state queries.

- Answer is phrased naturally and consistent with image-grounded behavior for its Level.

### Output Return exactly one token: PASS or FAIL. No explanations.

### Item to Evaluate - Level: {Level} - Question: {Question} - Question: {Question}

### Your Evaluation:

---

**Prompt for Filtering Stage 2**

### Role You are the BEST-FIT VALIDATOR. Decide ONLY whether the assigned Level is the BEST FIT among A/B/C/D for the given (Question, Answer).

If ANY other Level fits better than the assigned Level, output FAIL. Do NOT relabel or suggest a new Level.

### Canonical Level definitions (for comparison only)

Level 0: no ambiguous terms; single clear interpretation; direct factual answer.

Level 1: ambiguous term present; exactly one dominant target; answer explicitly clarifies what the ambiguous term refers to and then provides the definitive descriptive answer in a natural full sentence.

Level 2: ambiguous with 2–3 plausible targets; answer enumerates ALL in natural descriptive sentences (no single best-guess).

Level 3: ambiguous with many/equally plausible targets; no reasonable best guess; answer politely requests clarification without listing options and briefly states why clarification is needed.

### Best-Fit Priority Rules

- If no ambiguous term → prefer 0.

- If ambiguous term and one dominant target → prefer 1.

- If 2–3 plausible targets and the answer lists all → prefer 2.

- If many/equally plausible targets and the answer requests clarification (no listing) → prefer 3.

- If multiple seem possible, choose the most specific by these rules.

### Task - PASS iff the assigned Level is the unique best fit.

- FAIL if any other Level appears more appropriate or equally/more consistent.

### Universal Sanity Checks (must hold; otherwise FAIL)

- Question about observable visual properties only.

- No speculative/hidden-state queries.

- Answer phrased naturally and consistent with image-grounded behavior.

### Output Return exactly one token: PASS or FAIL.

### Item to Evaluate

- Assigned Level: Level

- Question: Question

- Answer: Answer

### Your Evaluation:

---

**Prompt for Filtering Stage 3**

### Role

You are the QUALITY VALIDATOR. Decide ONLY whether the (Image, Question, Answer) is suitable for a real-world, image-grounded VQA dataset.

Do NOT re-evaluate or change the assigned Level. Fail on quality issues only.

### Pass Conditions (ALL must hold)

REAL-WORLD IMAGE:

- Single real-world photograph (not drawing/CGI/collage/split-screen/UI screenshot).

- Sufficient visual information: not blank, not extremely blurred/overexposed/too dark, not extreme crop.

OBSERVABLE-ONLY QUESTION:

- About clearly visible, observable properties (color, shape, size, position, count, visible text, obvious material).

- No hidden states, internal contents, functionality, temperature; no brands unless clearly visible.

IMAGE-GROUNDED ANSWER:

- Consistent with what can be seen; no external knowledge.

- Wording aligns with the assigned Level's expected behavior (0=definitive; 1=confident; 2=lists;3=clarifies). (Do NOT reclassify—only fail if clearly contradictory.)

BASIC SANITY:

- Q–A refers to elements plausibly in-frame; no off-frame references.

- No unreadable OCR claims; no unsafe/privacy-violating content.

### Fail Conditions (ANY → FAIL)

- Non–real-world/composite image; insufficient visual info.

- Question requires inference beyond visible evidence.

- Answer contradicts image-groundedness or Level's behavioral form.

- References to objects/text not legibly present.

### Output

Return exactly one token: PASS or FAIL.

### Item to Evaluate

- Assigned Level: Level

- Question: Question

- Answer: Answer

### Your Evaluation:

---

**Prompt for Factual Consistency**

### Instruction:

You are a judge evaluating whether a model's answer is factually correct given the image.

- Ignore style, grammar, or completeness.

- PASS if the answer correctly refers to something that is actually visible in the image, even if it does not mention everything or omits other details.

- FAIL only if the answer mentions something not present in the image, contradicts the image, or hallucinates details.

### Output:

Return exactly one label:

- PASS

- FAIL

### Visual Question and Answer:

- Question: {Question}

- Answer: {Answer}

### Response:

---

**Prompt for Strategic Accuracy**

### Instruction:

You are an LLM-as-judge evaluating how a model's answer handles ambiguity.

Ignore the question completely. Judge only the style and content of the answer.

Classify the answer as Level 0, 1, 2, 3, or 4 according to the definitions below.

Do not provide explanations or reasoning — output only the type label.

### Type Definitions:

* Level 0 – Clear VQA

- Answer is a direct, factual statement.

- No mention of resolving ambiguous terms.

- Reads as a straightforward description of what is visible.

* Level 1 – Context-resolved Ambiguity

- Answer explicitly clarifies what an ambiguous term refers to, then provides the definitive factual description.

- Typically has a two-part structure: clarification of the referent + final descriptive answer.

- Must be a natural full sentence, not just a fragment.

* Level 2 – List All Plausible Options

- Answer enumerates 2–3 distinct possibilities.

- Each possibility is described in natural sentences (not bullet points).

- No single option is selected as the "best guess".

* Level 3 – Clarification Required

- Answer does not attempt to guess or enumerate.

- Politely requests clarification from the user.

- Acknowledges that multiple possibilities exist without listing them.

- Briefly explains why clarification is needed.

* Level 4 – Unknown

- The answer does not fit any of the above patterns.

- Use this if the answer is irrelevant, nonsensical, off-topic, or mixes multiple types in a way that does not clearly align.

### Answer:

- {Answer}

### Response:

---

**Prompt for Strategy-Prompting**

###Instruction:

Look at the image and the question, and respond strategically based on the level of ambiguity.

- If there is no ambiguity, answer clearly and factually.

- If the question uses an ambiguous term but context makes one target obvious, first clarify what the ambiguous term refers to, then provide the definitive factual answer in a natural full sentence.

- If the question allows two or three plausible targets, describe all of them in full sentences without choosing a single best guess.

- If the question has too many or equally plausible targets, politely ask for clarification.

###Question:

{question}

###Response:

---

**Prompt for Clarification Subset**

### Instruction:

You are a data constructor for Visual Question Answering (VQA).

Given (1) an ambiguous question about an image and (2) a clarification response, generate a resolved annotation in JSON format.

### TASK:

Your output must include:

- attr_type: the attribute type of the question (choose from: color, shape, position, count, visible_text, material, etc.)

- Hint: one sentence that uniquely identifies the target object in the image

- Q_resolved: the clarified sentence (not question type), rewritten to match the resolved meaning while keeping the same attribute type

- A_gold: a confident, single-sentence answer grounded in the image (no hedging or uncertainty)

### CONSTRAINTS: - attr_type must be exactly one of the listed categories.

- Hint must uniquely describe the object using clear visual cues (category, position, relations, or visible text).

- Q_resolved must stay aligned with attr_type.

- A_gold must be one confident sentence, no ambiguity, no hedging.

- Output valid minified JSON only.

### INPUT:

Ambiguous Question: {Question}

Clarification Response: {Response}

### SCHEMA: {"attr_type":"...","Hint":"...","Q_resolved":"...","A_gold":"..."}

### Response:

