# OpenReview forum: "AQuA: Toward Strategic Response Generation for Ambiguous Visual Questions"
_ICLR.cc/2026/Conference — ICLR 2026 Poster_

### Official Review · Reviewer_fMCK · 2025-10-23

**Soundness:** 3
**Presentation:** 3
**Contribution:** 3
**Rating:** 8
**Confidence:** 4

**Summary:**

The paper proposes a new benchmark dataset for training and evaluating Vision–Language Models (VLMs), specifically on how to handle different types and degrees of ambiguity in visual questions. The differentiating idea is to extend the binary abstention decision to a four-level taxonomy of ambiguity. Also, the dataset is labelled with four optimal response strategies: direct answer, context-based inference, enumerating plausible alternatives, or explicit clarification.

**Strengths:**

- The four-level taxonomy of ambiguity is very reasonable and novel.
- The dataset is well structured. The label validation is thorough. Overall, solid methodology-wise.
- The dataset also aligns well with two-stage training, with SFT and GRPO subsets.
- The failure mode analysis is informative.

**Weaknesses:**

- Like other datasets derived from existing labels and GPT models, there would be potential biases. More discussion on bias mitigation would be helpful.
- The four-level taxonomy is better than binary decisions. But it can still be too rigid. More experiments and analysis on ambiguous cases would be helpful.

**Questions:**

How does the trained model perform on non-COCO datasets?

---

> ### Author Response · Authors · 2025-11-21
> **Response to Reviewer fMCK**
>
> We appreciate your thorough review and feedback on our paper.
>
> **(1) Dataset generation with GPT**
>
> As you noted, our AQuA dataset is synthetic and generated using GPT. However, as described in Section 3.3, we conduct human verification via Amazon MTurk for all examples in the test split. This allows us to ensure quality without relying solely on GPT outputs and to better align the dataset with human judgments.
>
> Also, the authors were directly involved throughout the data generation pipeline, including refining the generation procedure, filtering low-quality samples, and validating the final outputs. These additional stages of human oversight reduce unintended biases that could arise from relying exclusively on GPT-generated content. While no synthetic dataset can be completely free of bias, our multi-stage process, with both human intervention and crowd-sourced validation, substantially mitigates such risks.
>
> We hope this additional clarification addresses your concerns regarding bias mitigation.
>
> **(2) Further analysis on more cases**
>
> Thank you for your valuable comment. To clearly distinguish different degrees of ambiguity, our taxonomy necessarily uses discrete levels. However, as the confusion matrix in Figure 5 indicates, the boundaries between levels naturally exhibit some gray areas rather than being perfectly rigid.
>
> We initially designed our four-level taxonomy heuristically based on internal experiments. In the review process, we conducted a large-scale human study to examine how people actually make strategic choices under ambiguous settings (please refer to **General Response 2**). In this evaluation, we presented workers with the possible strategy options and asked them to select the one they believed to be most appropriate. The results show that human choices closely align with our level design, while also revealing the expected gray zones in borderline cases. In particular, the failure cases are concentrated between Levels 2 and 3, where annotators differ in how they judge these borderline scenarios. We provide several qualitative examples of such cases in lines 414–450.
>
> Although these gray zones naturally arise in human decision-making, implementing and training a model requires discrete supervision. For this reason, we adopted a rigid level assignment when constructing AQuA, even while recognizing that some borderline cases exist. We will clarify this in the revised version.
>
> **(3) Evaluation on non-COCO dataset**
>
> Thank you for your thoughtful question. We generated our additional test samples using the Open Images V7 dataset and performed additional experiments. The results of these experiments are included in the **General Response (1)**.

---

> > ### Author Response · Authors · 2025-11-26
> > **Remind to Reviewer fMCK**
> >
> > **Dear Reviewer fMCK,**
> >
> > We appreciate your effort in reviewing our paper and understand your busy schedule. We have carefully addressed all of your concerns and questions in our response. If anything remains unclear, we would be glad to provide further clarification. Also, if our responses address your concerns, we would appreciate it if you could kindly consider adjusting the score. Thank you again for your time and invaluable feedback.

---

### Official Review · Reviewer_D7BQ · 2025-10-26

**Soundness:** 3
**Presentation:** 3
**Contribution:** 3
**Rating:** 6
**Confidence:** 3

**Summary:**

The paper introduces AQUA, a dataset trying to solve the ambiguity issues in VQA. There are four fine-grained levels. This work fine-tunes VLMs with SFT followed by GRPO using an LLM-as-a-judge reward. Fine-tuning small models substantially improves the accuracy across ambiguity levels, outperforming larger open- and closed-source baselines.

**Strengths:**

1. The research problem is clearly framed, with 4 levels of categorization
2. The dataset construction pipeline has human validation

**Weaknesses:**

1. The importance of the problem in real-world settings. In Figure 1, the other models' answers still seem reasonable to me. So I wonder about the significance of the problem in the VQA setting.
2. The rationale/completeness behind the 4 different levels. How can you tell whether there aren't other ambiguous questions?
3. It seems that the difference between the levels is simply the number of salient objects, which can be quite subjective or prone to errors. You need to pre-define a size threshold, which seems arbitrary.

**Questions:**

1. Since LLMs can be used a a judge for reward assignment, one question is, why cannot this ambiguity problem solved via prompting techniques? If GPT-5-mini can serve as the judge, can gpt-5-mini, with the proper prompts, just directly solve the ambiguous VQA problem? Is there really a need for fine-tuning? E.g., make this iterative -- let one LLM first give an answer, then use gpt-5-mini judge it, then iteratively let the LLM refine its answer. What will be the results of this approach? What's the trade-off here?

---

> ### Author Response · Authors · 2025-11-21
> **Response to Reviewer D7BQ (1/2)**
>
> Thank you for taking the time to review our paper and provide valuable feedback.
>
> **(1) About Figure 1**
>
> In Figure 1, there are multiple baseball bats, and none of them is particularly salient. Qwen responds as if there were only a single bat, leading to an entirely incorrect answer. GPT and Gemini refer only to the bat in the foreground, implicitly treating it as the intended target. In contrast, our model requests clarification about which bat the question refers to. Because the question-image pair is ambiguous, and no specific bat stands out as salient, we consider that requesting clarification is the more strategically appropriate response.
>
> **(2) Four level definition**
>
> Thank you for your question. In this work, we focus on situations where the presence of multiple objects and an underspecified question make it unclear which object should be the target. Our four-level taxonomy is designed to capture this specific type of visually grounded ambiguity, where the number and saliency of plausible target objects meaningfully determine which response strategy is most appropriate.
>
> We acknowledge that other kinds of ambiguity can arise in real-world settings. However, many of these do not naturally support a graded, level-based structure or the type of strategic variation addressed in our study. In such cases, the most reasonable behavior is often to request clarification immediately, rather than applying a nuanced ambiguity level. For this reason, we restrict our scope to object-referential ambiguity in multi-object scenes, an area where defining levels and corresponding strategies is both meaningful and actionable. We hope this clarifies the rationale and scope behind our four ambiguity levels.
>
> **(3) Salient objects**
>
> In this work, our focus is on whether VLMs can effectively handle ambiguity that arises when multiple objects could plausibly be the target and the question does not specify which one. Because ambiguity naturally increases as the number of plausible targets increases, our taxonomy organizes response strategies around this variation rather than treating it as a purely binary decision.
>
> Regarding dataset construction, the brief description in line 194 only summarized the idea of saliency; our actual procedure is more systematic. For each object, we compute a saliency score based on a weighted combination of its area ratio (0.7) and its distance from the image center (0.3), and identify the highest-scoring object. Through empirical analysis, we found that objects with a score above 0.6 can be reliably treated as salient.
>
> Also, we conducted a human study to examine how human actually make strategic choices under ambiguous scenarios, and this analysis revealed an important observation related to saliency (please refer to **General Response 2**). Human annotators performed almost perfectly on Level 1 examples (48 out of 50 correct), indicating that they were able to consistently identify the salient object in these single-salient-target settings. This provides additional evidence that our saliency definition is well aligned with how humans perceive visual prominence in practice.
>
> We apologize for any confusion and will provide a clearer explanation of this in the revised version.

---

> > ### Author Response · Authors · 2025-11-21
> > **Response to Reviewer D7BQ (2/2)**
> >
> > **(4) Prompting with GPT vs. Fine-tuning**
> >
> > Thank you for your valuable question. While we use GPT models during dataset generation and reward computation, directly solving ambiguous VQA problems through prompting alone remains highly challenging. **It is also important to note that when GPT is used for data generation in our pipeline, the process involves multiple carefully designed stages as well as substantial human verification to ensure quality; this is far more complex than simply asking GPT to answer the ambiguous questions directly.** To examine this, Table 1 reports the performance of GPT-5 under several prompting settings: zero-shot, Chain-of-Thought (CoT), and Strategy Prompting, where we explicitly provide our ambiguity-handling strategies in the instruction. As shown, GPT-5 still has difficulty reliably resolving ambiguity. Strategy Prompting helps to a limited extent, but several levels remain unsolved, and the overall performance is still notably below that of our fine-tuned model.
> >
> > In addition, GPT-5’s role as a reward model in our pipeline is fundamentally different from generating answers. The reward model only checks whether a response matches our predefined strategy categories, whereas generating an appropriate response under ambiguity requires much more reasoning. Our data generation pipeline also involves multiple stages with human verification, making it far more complex than simply prompting GPT-5 to answer the questions directly.
> >
> > Following your suggestion, we also experimented with an iterative prompting setup to test whether GPT-5 can progressively refine its answer through repeated judging and correction. For this analysis, we randomly sampled 50 test examples from each level (200 examples in total). In each iteration, we prompted GPT-5 with our strategy guidelines, let it generate an answer, used GPT-5-mini to judge whether the strategy was correct, informed GPT-5 of any mistakes, and asked it to revise its response. We repeated this process for 10 iterations.
> >
> > |  | **Level 0** | **Level 1** | **Level 2** | **Level 3** | **Overall** | **Unk** |
> > | --- | --- | --- | --- | --- | --- | --- |
> > | Stage 1 | 95.0 | 60.0 | 11.0 | 3.0 | 42.25 | 1 |
> > | Stage 2 | 97.0 | 77.0 | 14.0 | 8.0 | 49.0 | 5 |
> > | Stage 3 | 98.0 | 83.0 | 19.0 | 15.0 | 53.75 | 3 |
> > | Stage 4 | 97.0 | 89.0 | 25.0 | 16.0 | 56.75 | 6 |
> > | Stage 5 | 98.0 | 91.0 | 34.0 | 14.0 | 59.25 | 11 |
> > | Stage 6 | 99.0 | 91.0 | 35.0 | 16.0 | 60.25 | 7 |
> > | Stage 7 | 98.0 | 94.0 | 41.0 | 17.0 | 62.5 | 5 |
> > | Stage 8 | 100.0 | 97.0 | 39.0 | 16.0 | 63.0 | 2 |
> > | Stage 9 | 100.0 | 96.0 | 39.0 | 20.0 | 63.75 | 7 |
> > | Stage 10 | 100.0 | 95.0 | 43.0 | 19.0 | 64.25 | 6 |
> >
> > As shown in the table above, iterative refinement yields modest improvements across stages, but **GPT-5 continues to struggle on highly ambiguous cases**. Moreover, this iterative loop introduces substantial computational and latency costs, making it impractical as a general solution. **Even with multiple rounds of self-correction, GPT-5 does not reach the level of consistency achieved by a fine-tuned model.**
> >
> > These findings reinforce that prompting alone, even when iterative, is insufficient for robust ambiguity handling, and fine-tuning remains necessary to achieve consistent, level-aware behavior. We will update these insights in the revised version.

---

> > > ### Author Response · Authors · 2025-11-26
> > > **Remind to Reviewer D7BQ**
> > >
> > > **Dear Reviewer D7BQ,**
> > >
> > > We appreciate your effort in reviewing our paper and understand your busy schedule. We have carefully addressed all of your concerns and questions in our response. If anything remains unclear, we would be glad to provide further clarification. Also, if our responses address your concerns, we would appreciate it if you could kindly consider adjusting the score. Thank you again for your time and invaluable feedback.

---

### Official Review · Reviewer_tdiC · 2025-11-01

**Soundness:** 3
**Presentation:** 3
**Contribution:** 2
**Rating:** 4
**Confidence:** 3

**Summary:**

This paper tackles an interesting problem: teaching VLMs to handle ambiguous questions through strategic responses across four ambiguity levels. The motivation is solid and the experiments are thorough, but I have concerns about the heavy reliance on GPT-5 throughout the entire pipeline, the small dataset size, and some questionable design choices. The core idea has merit, but the execution has limitations that affect how much we can trust the results.

**Strengths:**

- The core idea is well-motivated
- Getting 3B models to outperform 72B+ models shows this training approach works.

**Weaknesses:**

- Generation, filtering, and evaluation all use GPT-5 variants. This creates circular logic—you're essentially teaching models to mimic GPT-5's behavior and then using GPT-5 to judge success
- 3.6K training samples from COCO only. Will this generalize to other domains?
- Why 20% bounding box area for Level 1? Why not 15% or 25%? No ablation studies to justify these choices.
- How do humans perform on strategic selection?
- Performance drops from 92.22% to 77.0% (Fig. 5). The "redistribution" explanation feels unsatisfying—this is a big drop.

**Questions:**

Please answer the questions in the weaknesses section

---

> ### Author Response · Authors · 2025-11-21
> **Response to Reviewer tdiC**
>
> We appreciate your insightful feedback on our paper.
>
> **(1) Usage of GPT-5**
>
> While we use GPT-5 variants during dataset generation and reward modeling, our pipeline is not simply replicating GPT behavior. To ensure quality and alignment with human preferences, **we include a human verification step during dataset generation.** Furthermore, as detailed in Sections 3.2 and 3.3, the generation process consists of several carefully designed stages, making it far more complex than a direct generate-filtering-evaluate loop with GPT.
>
> Importantly, Table 1 already shows that GPT-5 itself struggles to handle ambiguity when asked to perform the task directly. Even when we explicitly provide the response strategies through instruction design, GPT-5 often fails to produce correct or level-consistent answers. This indicates that the behavior learned through fine-tuning goes beyond what GPT-5 can reliably achieve through prompting alone, rather than simply mirroring GPT-5’s own outputs.
>
> **(2) Generalize to other domains**
>
> Thank you for your valuable feedback. To address your question, we created additional samples based on a different dataset (not COCO) and report the corresponding results in **General Response (1)**. We hope this explanation addresses your concerns and help in your final decision.
>
> **(3) Object for Level 1**
>
> Thank you for your thoughtful comment, and we apologize for any confusion. Regarding dataset generation, the brief mention in line 194 was intended only as an simple example, and it did not describe the full method we actually use to compute object choice. In practice, our method is more structured. We assign each object a saliency score computed from a weighted combination of its area ratio (0.7) and its distance from the image center (0.3). The object with the highest score is treated as the most likely candidate. Through empirical analysis, we found that objects with a score above 0.6 can be reliably considered salient.
>
> In addition, as discussed in response (4), we conducted a human evaluation to examine how people make strategic choices under ambiguity. **Notably, human annotators achieved a very high pass rate on Level 1 examples, indicating that they consistently identified the same salient object that our scoring method selects.** This alignment suggests that our saliency definition captures human intuitions about visual prominence reasonably well.  We will clarify this scoring process in greater detail in the revised version of the paper. We hope this explanation addresses and resolves your concerns.
>
> **(4) Human evaluation on strategic selection**
>
> Thank you very much for the valuable suggestion. We provide a detailed explanation in **General Response (2)**, and we sincerely hope this helps address your concern and informs your final decision.
>
> **(5) Performance drop on Level 1**
>
> Thank you for raising this important concern. As discussed in lines 377–403 and shown in Figure 5, the performance drop after GRPO is largely driven by the correction of strategy biases that were reinforced during SFT. Rather than learning the intended strategy distinctions, the SFT model tends to default overwhelmingly to Level 1 responses. This can be seen in the confusion matrix in Figure 5: many failure cases collapse into Level 1, indicating that the model has overfitted to this dominant strategy rather than genuinely understanding the underlying ambiguity levels.
>
> GRPO adjusts these misaligned preferences by encouraging the model to adopt strategies that better match the level-specific ambiguity in each example. As the model moves away from this overuse of Level 1 and begins applying higher-level strategies more appropriately, its performance, measured under the SFT-biased distribution, naturally decreases. Importantly, this drop does not reflect a degradation in the model’s reasoning ability. Instead, it reflects a shift away from the overly frequent (and often incorrect) use of Level 1 and toward behavior that aligns more closely with the intended strategy definitions. We will revise this point to clarify this interpretation more explicitly.

---

> > ### Author Response · Authors · 2025-11-26
> > **Remind to Reviewer tdiC**
> >
> > **Dear Reviewer tdiC,**
> >
> > We appreciate your effort in reviewing our paper and understand your busy schedule. We have carefully addressed all of your concerns and questions in our response. If anything remains unclear, we would be glad to provide further clarification. Also, if our responses address your concerns, we would appreciate it if you could kindly consider adjusting the score. Thank you again for your time and invaluable feedback.

---

### Official Review · Reviewer_xhCh · 2025-11-04

**Soundness:** 2
**Presentation:** 4
**Contribution:** 2
**Rating:** 4
**Confidence:** 4

**Summary:**

The paper introduces a new fine-grained dataset AQuA with ambiguous questions that requires vision-language models to recognize when they cannot answer a question. Most models answer ambiguous questions confidently instead of abstaining or seeking clarifications. The paper defines a 4-level categorization of question ambiguity, and an answering strategy for each level.  L0 being easy unambiguous qs, L1 qs are also unambiguous but require the model to resolve the salient referent, L2 qs have 2/3 possible answers, and L3 has qs where enumeration is inefficient and the model must request clarification.

AQuA is built using COCO images and uses provided bbox annotations to control ambiguity levels, for ex. L1 images have a single salient obj (exactly 1 bbox covering at least 20% of the image). GPT-5 is used to generate question-answer pairs for each level. The dataset is filtered to verify ambiguity level and answer correctness using GPT-5-mini (7.2K samples in final dset). The eval split is further filtered with human annotators who verify if each sample belongs to the assigned ambiguity level.

Experiments are performed on 4 open-source models (qwen2.5 & internvl3 family) and on GPT-5, Gemini-2.5-Flash.  Pretrained models are evaluated using zero-shot prompts, CoT prompts, and a strategy prompt. Furthermore, qwen2.5vl-3b & internvl3-2B models are finetuned on AQuA to handle ambiguous questions, first in a supervised manner followed by RFT using GRPO. In GRPO, a generation gets a reward (from GPT-5-mini as a judge) of 1 for grounded answer & correct strategy, and partial reward for correct strategy. Results are shown on the eval set of AQuA, where the trained models show better ability to choose the correct answer strategy.

**Strengths:**

- The paper identifies and attempts to tackle the issue of overconfident predictions by vision-language models for questions that are ambiguous.
- The data generation pipeline of AQuA is described in detail. Human filtering on eval split is performed to ensure clean samples.
- The paper is well written and easy to follow.

**Weaknesses:**

- The dataset is not meaningfully 'fine-grained'. There are only 4 categories of ambiguity, with real-world objects (also not from fine-grained categories)
- AQuA has a single fixed "correct" answer strategy for each level which is unrealistic. In real interactions multiple strategies (or combinations of them) are also appropriate. For instance AQuA says the only acceptable strategy for answering L3 questions is to ask for clarifications, whereas realistic answers could involve making a best-guess based on stated assumptions, followed by alternative coarse answers, followed by user clarifications etc.
- Samples are strictly classified into 4 categories, which is not reflective of real scenarios that can fall in between categories. Many cases lie between levels, for ex, an image with 5-10 apples falls in b/w L2 & L3, for which acceptable solutions can involve enumerating a few options and then asking for clarifications.
- Issue with metrics:
The *strategic accuracy* metric measures the ability of the model to conform to the categorizations made by AQuA and does not measure the true ability of the model to handle ambiguity. As discussed above, having a fixed strategy as ground-truth is unrealistic. The factual consistency prompt does not check for correctness of the answer and only measures groundedness. Better evaluation metrics are needed to measure a model's effectiveness for ambiguous questions (including checking correctness of answers).

**Questions:**

- In RFT, rewards are provided by GPT-5-mini. What is the computational overhead of this? How does training time compare to simpler alternatives like a locally hosted judge, or simpler format based rewards (for example looking for words similar to "clarify" in answers to L3 questions)?
- RFT is performed using just 60 training samples. Is there any merit to choosing more data?
- Performance on Out-of-Domain data. Are AQuA-trained models generalizable? Evaluation on OOD ambiguity datasets such as VQ-FocusAmbiguity[1], ClearVQA[2], and on hallucination benchmarks like POPE[3], AMBER[4], HaloQuest[5] would strengthen claims.
- Strategy Prompting seems effective in generating grounded responses. It would be interesting to see qualitative outputs of the same.

I believe the formulation of AQuA with strict 4-level taxonomy and a single "correct" strategy limits its practical utility (as mentioned in pts 2&3 in weaknesses). Furthermore, the strategic acc metric measures conformity to the proposed levels rather than measuring the model's ability to answer ambiguous queries. The factual acc metric only checks for groundedness and not for correctness of the answer. In light of these issues, I believe AQuA does not provide a practical, reliable way to quantify performance under ambiguity.


[1] Chen, C., Tseng, Y., Li, Z., Venkatesh, A., & Gurari, D. (2025). Acknowledging Focus Ambiguity in Visual Questions.
[2] Jian, Pu et al. “Teaching Vision-Language Models to Ask: Resolving Ambiguity in Visual Questions.” ArXiv abs/2507.13773 (2025): n. pag.
[3] Li, Yifan et al. “Evaluating Object Hallucination in Large Vision-Language Models.” Conference on Empirical Methods in Natural Language Processing (2023).
[4] Wang, Junyang et al. “An LLM-free Multi-dimensional Benchmark for MLLMs Hallucination Evaluation.” ArXiv abs/2311.07397 (2023): n. pag.
[5] Wang, Zhecan et al. “HaloQuest: A Visual Hallucination Dataset for Advancing Multimodal Reasoning.” ArXiv abs/2407.15680 (2024): n. pag.

---

> ### Author Response · Authors · 2025-11-21
> **Response to Reviewer xhCh (1/2)**
>
> Thank you for valuable reviews on our paper.
>
> **(1) Level definitions**
>
> Our goal in this work is to understand how VLMs should respond when multiple objects are present and the question is ambiguous about which one should be the focus. To study this, we define four levels based on the number of plausible target objects, with each level corresponding to the **most strategically effective** response behavior for that degree of ambiguity.
>
> We also examined how humans make strategic choices under same ambiguous scenarios. We found that human decision patterns align closely with our level definitions, particularly in how they judge when to answer directly, list all plausible options, or request clarification (please see **General Response 2**). These results support the validity of our four-level framework for characterizing ambiguity-handling strategies.
>
> We will include these human alignment findings in the revised version to further strengthen the motivation behind our level design.
>
> **(2) Fixed Strategy**
>
> Thank you for your comment. In our taxonomy, making a best-guess response is appropriate in Level 1 cases because there is a clearly salient object that the model can reasonably infer as the intended target. In contrast, Level 3 cases are defined precisely for situations where no object stands out as salient, and therefore a best-guess answer would not be reliable. This is why Level 3 emphasizes requesting clarification rather than guessing.
>
> In addition, for example, in the Level 1 instance from Figure 2, when we follow up with *"No, I mean coleslaw."* the model updates its answer to: *"The term 'this' refers to the side dish—coleslaw—and it's topped with shredded cabbage and carrots."* This demonstrates that **the model first makes a reasonable best guess** when a salient object is present, and **then adapts appropriately when the user clarifies their intent**. We observe similar adaptive behavior in Level 2 scenarios as well. We will include this finding in the revised version.
>
> **(3) Level 2 vs. Level 3**
>
> Our objective is to guide models toward the most effective strategy for responding under ambiguous conditions. When the context allows a clear inference or when there is a single salient object, the model should simply provide the answer. When multiple plausible candidates exist, the model may either enumerate them or ask for clarification. Although enumerating all candidates can sometimes be feasible, it can also be inefficient and may still miss certain items. In such situations, requesting clarification is often the more effective and reliable choice. Our taxonomy is designed to help models recognize when clarification is preferable to attempting exhaustive enumeration.
>
> **In defining the boundaries between levels, we initially relied on heuristic analysis during internal development. In the review process, we also conducted a large-scale human study to more rigorously examine how people make strategic choices under ambiguous scenarios (please see General Response 2).** This evaluation revealed clear gray areas in cases with 3–4 plausible objects, where annotators were split between enumerating and requesting clarification. For scenes with roughly 5–10 plausible targets, human annotators generally preferred asking for clarification over listing all possibilities.
>
> We will incorporate these observations and insights into the revised version.
>
> **(4) Metric for correctness**
>
> Thank you for your question. In this work, the correctness of the answer is not our primary focus. As noted in lines 74–76 and Section 4.1, our goal is to equip existing VLMs with effective strategies for handling ambiguity. The ability to produce a factually correct answer is determined by the underlying VLM itself. Our evaluation therefore focuses on whether the model adopts an appropriate ambiguity-handling strategy and whether its response remains grounded in the image.
>
> **(5) Computational overhead**
>
> We use GPT-5-mini as the reward model during GRPO training. In our experiments, this introduced only a small computational overhead, and it remained practical within our training pipeline. We initially explored heuristic, keyword-based reward rules, but we found that such approaches were too brittle and often misaligned with the intended strategy definitions. Simple heuristics tended to miss many valid strategy realizations, leading to inaccurate rewards and poorer overall learning dynamics.
>
> For these reasons, we chose GPT-5-mini for reward modeling: while slightly more expensive than heuristics, it provides significantly more reliable reward signals and enables the model to learn the intended ambiguity-handling behaviors more effectively.

---

> > ### Author Response · Authors · 2025-11-21
> > **Response to Reviewer xhCh (2/2)**
> >
> > **(6) More training samples**
> >
> > Thank you for the suggestion. In our original setup, we performed GRPO training with 60 samples (15 examples per level). Based on your feedback, we conducted an additional experiment using 120 training samples (30 examples per level).
> >
> > |  | **Level 0** | **Level 1** | **Level 2** | **Level 3** | **Overall** | **Unk** |
> > | --- | --- | --- | --- | --- | --- | --- |
> > | InternVL (60 samples) | 98.78 | 80.0 | 59.67 | 78.0 | 79.11 | 12 |
> > | InternVL (120 samples) | 96.0 | 90.3 | 57.89 | 85.33 | 82.39 | 29 |
> > | Qwen (60 samples) | 99.56 | 77 | 82.22 | 86.33 | 86.28 | 1 |
> > | Qwen (120 samples) | 99.44 | 86.44 | 73.33 | 92.11 | 87.83 | 2 |
> >
> > As shown in the table above, increasing the training data from 60 to 120 samples leads to higher performance. Although the degree of improvement varies across levels, the results suggest that GRPO can further compensate for the limitations of the SFT model when provided with more data. Our initial choice of 60 samples was motivated by efficiency, but we will include in the revised version the observation that additional data can yield further gains.
> >
> > **(7) Other datasets**
> >
> > Thank you for the suggestion. We attempted to conduct additional evaluations along the lines you proposed, however, VQ-FocusAmbiguity and ClearVQA are not publicly available. As noted earlier, the goal of this work is not to produce fully correct answers, but to equip models with effective strategies for handling ambiguity. For this reason, we do not report results on hallucination benchmarks. Moreover, benchmarks such as POPE, AMBER, and HaloQuest rely on evaluation protocols that are not directly compatible with our response format. For example, POPE assesses hallucination by checking whether the model answers “yes” or “no”, whereas our models do not follow such a binary response style. This makes direct comparison difficult. We hope this explanation clarifies our evaluation choices and addresses your concerns.
> >
> > **(8) Strategy Prompting**
> >
> > Thank you for the suggestion. As you noted, strategy prompting can help some models handle ambiguity more effectively. It tends to be more beneficial for larger models, and we observe particularly strong improvements with GPT-based models.
> >
> > > **Image**: http://images.cocodataset.org/train2017/000000152788.jpg
> > >
> > > **Question**: Where is this lamp placed?
> > >
> > > **zero-shot**: On the side (end) table next to the sofa by the window.
> > >
> > > **Prompt**: There are two lamps visible, so it’s ambiguous which you mean. The smaller lamp with a shade is placed on a side table next to the sofa by the window. The taller floor lamp stands in the corner behind the loveseat against the back wall.
> >
> > The example above shows a case where GPT does not handle the ambiguity well in a zero-shot setting but succeeds when strategy prompting is applied. In this scenario, there are two plausible candidates: the zero-shot model mentions only one of them, whereas strategy prompting allows the model to enumerate both appropriately. Although this approach can be helpful, we find that models still struggle with certain levels of ambiguity when relying solely on prompting. We will include additional qualitative examples in the revised version to further illustrate these effects.

---

> > > ### Author Response · Authors · 2025-11-26
> > > **Remind to Reviewer xhCh**
> > >
> > > **Dear Reviewer xhCh,**
> > >
> > > We appreciate your effort in reviewing our paper and understand your busy schedule. We have carefully addressed all of your concerns and questions in our response. If anything remains unclear, we would be glad to provide further clarification. Also, if our responses address your concerns, we would appreciate it if you could kindly consider adjusting the score. Thank you again for your time and invaluable feedback.

---

### Author Response · Authors · 2025-11-21
**General Response to All Reviewers**

Thank you for the valuable reviews on our paper. Below are our general responses to common questions.

**(1) non-COCO dataset sample**

Following the suggestions from Reviewer tdiC and fMCK, we constructed additional test samples using a dataset other than COCO. Specifically, we applied the same data generation procedure (including human filtering) to the Open Images V7 dataset and created 100 samples per level (total 400 samples).

| **Model** | **Level 0** | **Level 1** | **Level 2** | **Level 3** | **Overall** | **Unk** |
| --- | --- | --- | --- | --- | --- | --- |
| Qwen (zero-shot) | 98.0 | 2.0 | 1.0 | 0 | 25.25 | 2 |
| Qwen (SFT+GRPO) | 97.0 | 87.0 | 76.0 | 89.0 | 87.25 | 2 |
| InternVL (zero-shot) | 98.0 | 1.0 | 5.0 | 1.0 | 26.25 | 7 |
| InternVL (SFT+GRPO) | 96.0 | 86.0 | 55.0 | 77.0 | 78.5 | 2 |

As shown in the table above, the samples generated from Open Images exhibit performance trends similar to those of the COCO-based samples. This suggests that although our model was trained on COCO, it generalizes well to test data derived from other datasets. We will include this observation in the revised version.

**(2) Human evaluation**

To examine how humans perform on strategic selection, we randomly sampled 50 test examples from each level (200 in total) and conducted a human evaluation on Amazon MTurk. For each image–question pair, we collected responses from three independent workers. We provided workers with the following strategy definitions and asked them to select the strategy they believed was most appropriate for each image–question pair:

- **Strategy 1: Answer Directly** — Use this when the target is obvious OR can be inferred from the image context.
- **Strategy 2: List All Possibilities** — Use this when there are a small number of plausible targets. It is more efficient to describe all of them than to request clarification.
- **Strategy 3: Ask for Clarification** — Use this when there are many possible targets or the scene is too crowded. Additional information is strictly required.

These correspond to Level 0–1, Level 2, and Level 3 respectively. For Strategy 2, we intentionally did not impose a strict numerical threshold for what counts as "a small number", allowing workers to interpret it freely. **For each example, we collected responses from three independent workers and assigned the final human label using a majority vote.** Examples without majority agreement were treated as fail cases (only 13 cases lacked agreement among the three workers).

|  | **PASS** | **FAIL** |
| --- | --- | --- |
| Level 0 | 50 | 0 |
| Level 1 | 48 | 2 |
| Level 2 | 32 | 18 |
| Level 3 | 32 | 18 |

As shown in the table, Level 0 and 1 exhibit very high pass rate, while Levels 2 and 3 contain more failures.

**Upon analyzing these fail cases in more detail, we found two main patterns.** First, many failures occurred in situations where the image contained three or four plausible target objects. **Because we did not specify to workers how many plausible objects should trigger a request for clarification**, workers differed in whether they should list all plausible options or ask for clarification, leading to inconsistent strategy choices. These cases account for roughly 20 of the failures observed within Level 2 and 3, and they fall squarely within the boundary scenarios that our level definitions were designed to separate.

Second, in 8 fail cases, workers appeared to disagree on which object should be considered salient. **Since we did not tell workers which object we viewed as salient, nor did we provide any numerical guidance about object prominence,** some workers inferred a salient target from factors such as focus or visual prominence, while others did not. These cases further illustrate that saliency judgments can vary across annotators when the scene is ambiguous. **Nevertheless, for the examples we defined as Level 1, where the saliency is clear, annotators consistently agreed, resulting in high pass rates.**

Overall, these findings show that while humans are broadly aligned with our level definitions, the decision boundary between "enumerating possibilities" and "requesting clarification" can be subjective in borderline cases. We will incorporate these findings and insights into the revised version.

---

### Meta-Review · Area_Chair_vArg · 2026-01-07

**Summary:**

This paper received mixed reviews, primarily due to concerns about task formulation clarity and the reliance on GPT-family models throughout dataset construction and evaluation. After carefully reviewing the manuscript and the author's rebuttal, the AC finds that the work makes a meaningful and timely contribution. The paper might be the first to systematically define multi-level visual ambiguity and to study how models should select appropriate response strategies rather than optimize for answer correctness. Clarifications in the rebuttal effectively addressed misunderstandings regarding the level definitions, emphasizing that the task focuses on ambiguity handling in scenes with multiple plausible interpretations, which aligns with realistic human reasoning. While concerns were raised about potential circularity due to the use of GPT-based models in data generation and reward modeling, the AC believes this does not invalidate the core contribution, as the goal is not hallucination reduction or accuracy improvement but structured ambiguity resolution. Importantly, the additional human evaluation results provided during the rebuttal demonstrate high agreement between human ambiguity resolution behaviors and the proposed levels, offering independent support for the validity of the design. Although broader model diversity would further strengthen the study, the current evidence sufficiently supports the paper’s claims. Overall, the AC considers this work a valuable step toward ambiguity-aware vision-language research and recommends acceptance.

**Reviewer Concerns:**

-- outstanding --: Generation, filtering, and evaluation all use GPT-5 variants. This creates circular logic—you're essentially teaching models to mimic GPT-5's behavior and then using GPT-5 to judge success 3.6K training samples from COCO only. Will this generalize to other domains?

**Reviewer Scores:**

Regarding the comment on RFT, where rewards are provided by GPT-5-mini and the reviewer asked about the computational overhead and comparisons to simpler alternatives (e.g., a locally hosted judge or simpler format-based rewards), the authors explicitly provided additional experimental results addressing this question. Based on these results, the AC believes that the reviewer would likely have revised their score had they been able to participate fully in the discussion.

---

### Decision · Program_Chairs · 2026-01-26

Accept (Poster)